# Harnessing photoinduced electron transfer to optically determine protein sub-nanoscale atomic distances

Antonios Pantazis[1,2,3], Karin Westerberg[4], Thorsten Althoff[5], Jeff Abramson[5] & Riccardo Olcese[1,5]

Proteins possess a complex and dynamic structure, which is influenced by external signals and may change as they perform their biological functions. We present an optical approach, distance-encoding photoinduced electron transfer (DEPET), capable of the simultaneous study of protein structure and function. An alternative to FRET-based methods, DEPET is based on the quenching of small conjugated fluorophores by photoinduced electron transfer: a reaction that requires contact of the excited fluorophore with a suitable electron donor. This property allows DEPET to exhibit exceptional spatial and temporal resolution capabilities in the range pertinent to protein conformational change. We report the first implementation of DEPET on human large-conductance $K^+$ (BK) channels under voltage clamp. We describe conformational rearrangements underpinning BK channel sensitivity to electrical excitation, in conducting channels expressed in living cells. Finally, we validate DEPET in synthetic peptide length standards, to evaluate its accuracy in measuring sub- and near-nanometer intramolecular distances.

[1] Division of Molecular Medicine, Department of Anesthesiology & Perioperative Medicine, UCLA, Los Angeles, CA 90095, USA. [2] Division of Neurobiology, Department of Clinical and Experimental Medicine (IKE), Linköping University, Linköping 581 83, Sweden. [3] Wallenberg Center for Molecular Medicine, Linköping University, Linköping 581 83, Sweden. [4] Amgen, Thousand Oaks, CA 91320, USA. [5] Department of Physiology, UCLA, Los Angeles, CA 90095, USA. Correspondence and requests for materials should be addressed to A.P. (email: antonios.pantazis@liu.se) or to R.O. (email: rolcese@ucla.edu)

luorescence spectroscopy is a seminal approach in structural biology, allowing the determination of structural protein information under physiologically-relevant experimental conditions[1,2]. Most applications of fluorescence spectroscopy in structural biology are based on Förster resonance energy transfer (FRET) between fluorescence donor and acceptor protein adjuncts[3,4]. While widely used and continually refined[5,6], even current FRET-based methods are not altogether without limitations, which compromise their spatial or temporal resolution. One limitation in particular, is that FRET always provides distances between fluorescent donor and acceptor protein adjuncts. This is hardly a concern when determining inter-molecular distances in protein complexes; however, fluorescent adjunct distances diverge significantly from protein atom distances and orientations in the sub- and near-nanometer scale pertinent to protein structure and function. To address this and other limitations of FRET methods, we have developed a new optical approach based on an alternative mechanism of distance-dependent modulation of fluorescence: distance-encoding photoinduced electron transfer, DEPET. DEPET directly provides the distance between protein backbone ($C_\alpha$) and side-chain ($C_\beta$) atoms, and is therefore particularly suited to the precise determination of the protein structure and the fine conformational changes pertinent to protein function. We demonstrate the capability of DEPET in determining how membrane depolarization changes intramolecular distances and side-chain orientations in the human large-conductance potassium (BK) channel: the universal regulator of cellular excitability[7]. We also validate the accuracy of DEPET in gauging short distances, by measuring the length of rigid polyproline peptides in solution.

In lieu of resonance energy transfer, DEPET is based on photoinduced electron transfer (PET): a means to quench the emission of a fluorophore in the singlet excited state, upon contact with a molecule of appropriate electronegativity[8,9]. Near-nanometer distance-measuring capability using PET fluorescence quenching has been demonstrated using the TEMPO nitroxide radical as a quencher[10]. This quencher, attached to probes of variable length was used to quantify distances in the Shaker K$^+$ channel[11]. This study demonstrated the potential of PET-based fluorescence quenching in resolving short distances, in the ~2.7–5 nm range, with sub-nanometer resolution. Achieving even shorter distance measurement capability requires a smaller fluorescence quencher. Fortuitously for a structural biology approach, an efficient PET quencher of a variety of small labels used in fluorescence spectroscopy is the side-chain of the tryptophan amino acid[8,12–16]. In DEPET, we evaluate the quenching of a small, site-directed fluorescent probe by a nearby native or introduced Trp residue to extract distances in the Ångström (subnanometer) range. In fact, DEPET is uniquely capable of resolving relative side-chain orientations in a protein under physiologically-relevant conditions.

To extract distance information in DEPET, fluorophores of different length are used (Fig. 1): if a fluorophore is too short, it will not be quenched by the distal Trp; if its length is sufficient, more quenching will be observed; taken together, this information encodes the distance between the fluorescently-labeled site and the quenching Trp. This approach is reminiscent of seminal work on ion channel proteins, where the distance between the channel pore and a site elsewhere in the protein was evaluated by using site-directed tethered pore blockers of varying length[17]. Previous attempts have been made to extract distance information from Trp-quenched fluorophores, without yielding a quantified result[15].

To obtain quantified distance results, it is important to characterize the flexibility and range of motion of each fluorophore. We used molecular dynamics (MD) simulations, which yield a function (the Fluorophore-Distance-Quencher function, FDQ) correlating quenching probability and the distance between the $C_\alpha$ atoms of the fluorescently-labeled cysteine and the quenching Trp. Using the same approach, the distance between the Cys $C_\alpha$ and the Trp $C_\beta$ atoms can also be determined: this information is useful for evaluating the orientation of the Trp side-chain with respect to the fluorescently-labeled site.

In this work, we implement DEPET in three sites of the voltage-sensing domain of the human BK channel, obtaining the distance and side-chain orientation of a native Trp residue with respect to juxtaposed transmembrane helices S0, S1 and S2 in its resting and active states. These measurements are in good agreement with recent cryo-EM-resolved structures of the BK channel. We further evaluate the accuracy of DEPET in polyproline length standards, predicting their length within, on average, 1.3 Å.

## Results

**Evaluating the quenching of rhodamine fluorophores by Trp.** PET-based Trp quenching of small fluorescent molecules involves brief collisional quenching and longer-lasting, static quenching, likely due to the formation of stable hydrophobic complexes[8,13]. Both mechanisms require van der Waals overlap between the Trp indole side-chain and the fluorescent moiety, i.e., contact; and both occur within the nanosecond time domain, rendering them indistinguishable in steady-state fluorescence measurements, such as those in DEPET. An important prerequisite is to establish whether Trp is equally efficient at quenching the fluorophores used, by evaluating its steady-state Stein-Volmer bimolecular quenching constant ($K_{SV}$)[1]. Indeed, the quenching efficiency of Trp for the tetramethylrhodamine (TMRM) fluorophores used in this study was determined to be very similar, with $K_{SV}$ ranging from 36 to 41 M$^{-1}$ (Supplementary Figure 1). This is not surprising considering that the fluorophores used have the same xanthene fluorescent moiety, and are thus spectrally identical (Supplementary Figure 1). When different fluorescent moieties are used, establishing their Trp $K_{SV}$ will be very useful, to normalize their FDQ functions and thus eliminate any difference in Trp quenching efficiency from the distance calculation.

**Implementing DEPET in the BK voltage sensor.** BK channels are membrane-spanning proteins that confer a K$^+$ conductance in response to two physiological signals: intracellular [Ca$^{2+}$] elevation and/or membrane depolarization[18,19]. That is, BK channels integrate electrical and biochemical signaling to regulate cellular excitability in diverse tissues, such as central neurons, endocrine cells, and smooth muscle[7]. Recent structures of a BK channel resolved at near-atomic resolution by cryo-EM demonstrate how metal ligands open this channel[20,21]. However, its mechanism of voltage-dependent activation is still unclear, as the voltage sensors in the unliganded/shut and liganded/open channel structures were resolved practically in the same conformation (Supplementary Figure 2).

Since the voltage sensors of this protein require strong membrane depolarization to activate[22], we investigated their structural dynamics by implementing DEPET in a voltage clamp context. Human BK channels were expressed in *Xenopus* oocytes and labeled at position 136, at the extracellular flank of transmembrane helix S1. We used a cut-open oocyte Vaseline gap apparatus[23,24] with epifluorescence capability[25,26] to simultaneously (i) control the membrane potential, (ii) track channel opening (K$^+$ current), and (iii) observe the fluorescence emission from S1-labeling fluorophores. We previously demonstrated that fluorophores labeling S1, and other BK transmembrane helices, exhibit state-dependent quenching by a native Trp residue

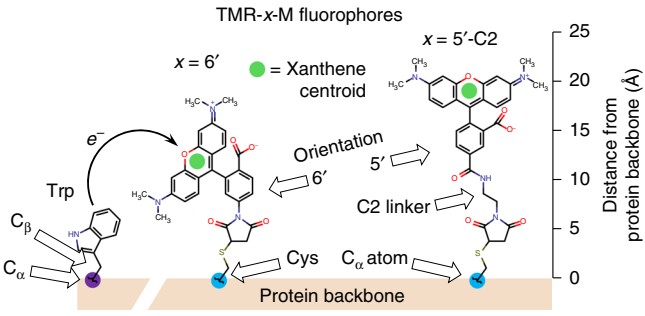

**Fig. 1** Principle of distance-encoding photoinduced electron transfer (DEPET). A tryptophan (Trp) residue is shown attached to the protein backbone, along with two Cys-conjugated tetramethylrhodamine maleimide (TMRM) fluorophores of different length. Trp residues are potent quenchers of TMRM fluorescence (Supplementary Figure 1 and ref. [16]), by the photoinduced electron transfer (PET) process: while the TMRM fluorescent moiety (xanthene, green circle) is in the excited singlet state, contacting a Trp side-chain triggers an electron transfer reaction, preventing fluorescence emission[8,9]. In this work, we show how we can extract distances and distance changes directly associable with protein structure and function (Cys $C_\alpha$ (cyan circle)—Trp $C_\alpha$ (purple circle) and Cys $C_\alpha$—Trp $C_\beta$ (red circle) atoms, respectively), by measuring the Trp-induced PET quenching of TMRM fluorophores of different length. We have implemented DEPET in (i) the activation transition of the human BK channel voltage-sensing domain, in conducting channels expressed in a cell and (ii) synthetic polyproline peptide length standards in solution

(W203) at the extracellular flank of helix S4[16,27,28]. In fact, when short TMR-6′-M label was used, its fluorescence increased upon depolarization, indicating less quenching in the Active state (Fig. 2a). This is not surprising since helix S4 possesses charged residues and is expected to undergo voltage-evoked conformational changes upon depolarization[29–31], as in other ion channels and even enzymes coupled to conserved voltage-sensing domains[32–36]. Accordingly, removal of W203 by site-directed mutagenesis practically abolished the observed fluorescence deflections (Fig. 2b). A straightforward structural interpretation of this result is that the S1 fluorophore is more quenched in the Resting conformation of the BK voltage sensor (negative membrane potential) than the Active state (positive membrane potential); therefore the distance between S1 (fluorophore) and S4 (Trp) increases upon activation.

When TMRM fluorophores of increasing length were used, the fluorescence changes progressively diminished (Fig. 2a). An interpretation of this result is that, while shorter labels experience differential quenching in the Resting and Active states of the voltage sensor, longer labels are presumably equally quenched by Trp in both states, reporting less voltage-dependent fluorescence change (Fig. 2c). The fluorescence deflections were normalized by macroscopic channel conductance to account for small variations in channel expression (Supplementary Figure 3a–f); all data from this experiment are shown in Fig. S3g. In addition, we performed control experiments mixing Cys-less with [203]Trp-less BK channel subunits, to exclude the possibility of inter-subunit fluorophore quenching (Supplementary Figure 4). We describe how the optical signals from S1-labeled channels in the presence of W203, acquired simultaneously with a measure of protein function (BK channel currents, Fig. 2a & Supplementary Figure 3a), can be converted to atomic distance and orientation measurements, below.

We also probed the voltage-dependent rearrangement of voltage-sensing helix S4 with respect to helices S0 and S2, by labeling positions 19 and 145, respectively (Supplementary Figures 5 & 6). The proportions of the voltage-evoked $\Delta F$

reported from the extracellular flank of S0 from TMRM fluorophores of different length were unlike those reported from helix S1 (compare Supplementary Figure 5a and Fig. 2a), hinting that the S0–S4 distance is different than S1–S4. On contrast, the shortest TMRM fluorophore reported the strongest $\Delta F$ when labeling helix S2, with diminished signals reported from fluorophores of increasing length (Supplementary Figure 6a), similar to the $\Delta F$ proportions of S1-labeling probes. As before, experiments in the absence of W203 were performed to confirm that the $\Delta F$ in this labeling position was mainly due to the interaction of the S0-conjugated or S2-conjugated labels with the S4 Trp (Supplementary Figures 5b & 6b).

**Determining distance and orientation**. The major premise of DEPET is that the differential state-dependent quenching of labels by a nearby Trp residue is due to state-dependent distance changes. To quantify the distances from the DEPET signals, a function is required to map Trp-induced quenching probability to the separating distance between the fluorophore attachment site (the $C_\alpha$ atom of the labeled Cys) and the Trp $C_\alpha$ (Fig. 1). To construct this function, the flexibility and range of movement of each fluorophore needs to be taken into account. This was achieved using MD simulations of each TMRM fluorophore conjugated to a Cys residue. The simulations generated possible conformers of the Cys-TMRM conjugate species (Fig. 3a); in turn, they were used to determine the frequency of encountering the xanthene fluorescent moiety a given distance from the Cys $C_\alpha$; i.e., a measure of fluorophore range of movement (Fig. 3b, green). The same exercise was performed for Trp; as a more constrained molecule, it yielded a much steeper function for the chance to encounter its side-chain (indole) a given distance from its $C_\alpha$ atom (Fig. 3b, gray). The interception of the two probabilities (chance to encounter xanthene ∩ chance to encounter indole) effectively describes the quenching mechanism; its calculation over a separating distance between the Cys and Trp $C_\alpha$ atoms provides the required function, to correlate fluorescence quenching with a measure of distance (Fig. 3c). This exercise was performed for all available fluorophores (Supplementary Figure 7) and the resulting probability density histograms were fit to empirical combinations of exponential and Gaussian functions, to facilitate the fitting of fluorescence data, and their use by the wider scientific community (parameters in Supplementary Table 1). We refer to the resulting Fluorophore—Distance— Quencher functions as $FDQ_{\alpha\alpha}$. Super-imposing the $FDQ_{\alpha\alpha}$ functions on the cryo-EM-resolved structure of the BK channel shows that the range of quenching of the fluorophores used hardly exceeds the diameter of the voltage-sensing domain (Supplementary Figure 8), accounting for the lack of inter-subunit quenching (Supplementary Figure 4).

Using the same MD-derived information, it is possible to calculate fluorescence quenching as a function of the distance between the Cys $C_\alpha$ and the Trp $C_\beta$ atoms: $FDQ_{\alpha\beta}$ (Supplementary Figure 9). This distance is important to determine the orientation of the Trp side-chain with respect to the label site, and provided an important validation for the implementation of DEPET on the BK channel. $FDQ_{\alpha\beta}$ parameters are listed in Supplementary Table 2.

For each labeled position, the fluorescence data ($\Delta F/G$) from TMRM labels of increasing length were fit to the FDQ functions for each fluorophore simultaneously, to determine the distance between W203 (S4) $C_\alpha$ and $C_\beta$ atoms and the labeled Cys $C_\alpha$, in the Resting ($d_R$) and Active ($d_A$) states, producing well-constrained fits (Fig. 4). Accordingly, the S4 Trp is within ~6–8 Å of S2 and S1 in the Resting state, but diverges, upon membrane depolarization, to ~13–14 Å. S4 also diverges from S0

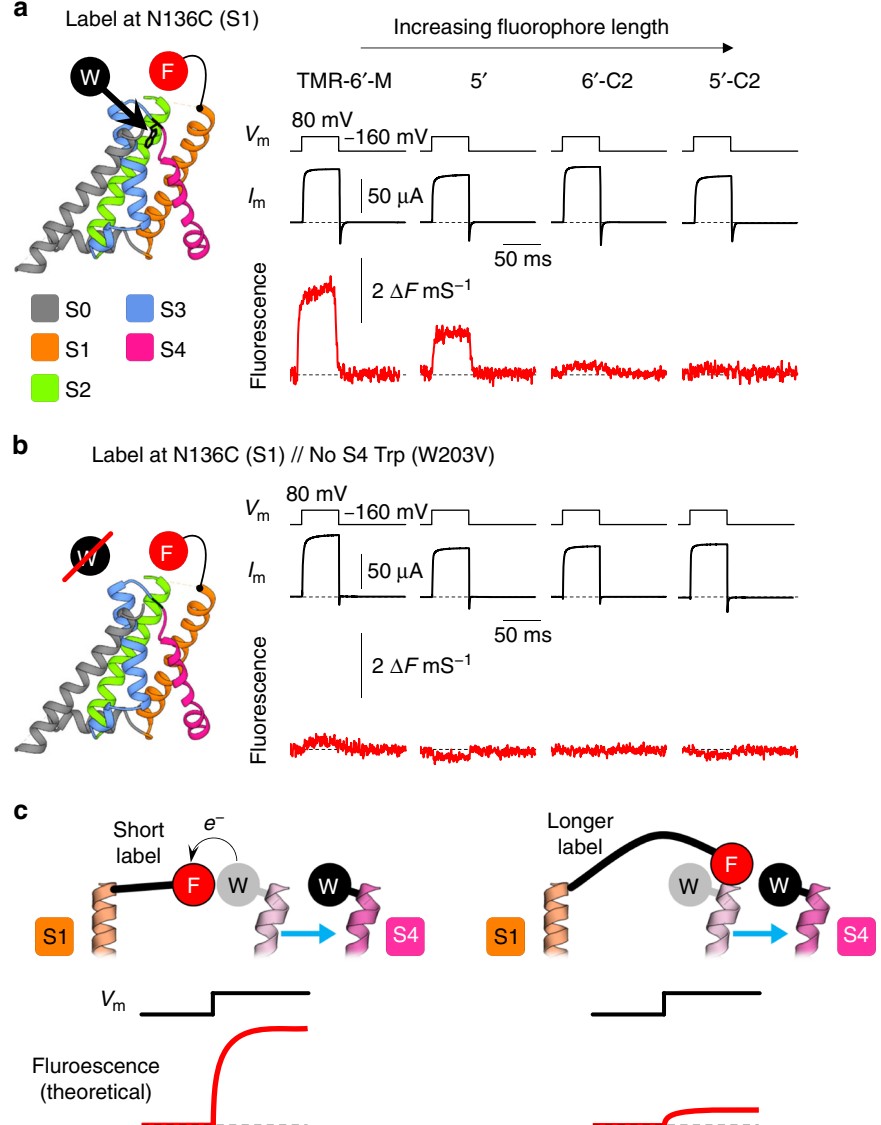

**Fig. 2** DEPET measurements in conducting, human BK channels. **a** Simultaneously-acquired $K^+$ current (black) and fluorescence (red) from oocytes expressing human BK channels fluorescently-labeled outside helix S1 (position 136) with different TMRM fluorophores, upon a 50-ms voltage pulse from $-160$ to $+80$ mV. Note that increasing fluorophore length, results in diminished amplitude of fluorescence deflections ($\Delta F$). **b** As above, when the native Trp extracellular to S4 in removed (W203V), $\Delta F$ is strongly diminished for the four shortest fluorophores, demonstrating that the fluorescence changes in W203 channels were due to the differential quenching of S2-bound TMRM fluorophores by the S4 Trp. All data for this experiment are shown in Supplementary Figure 3g. **c** A structural interpretation of the fluorescent signals: Left: when the BK voltage-sensing domain is in the resting state, the S4 Trp is near helix S1, quenching the fluorescent label attached to it; upon depolarization, S4 moves away, beyond the reach of the short S1 label: this molecular rearrangement is reported as fluorescence unquenching. Right: when a longer label is attached to S1, it is equally quenched by the S4 Trp in both Resting and Active conformations, so the same movement (S4 helix divergence from S1) is reported as a much fainter fluorescence deflection

during its voltage-dependent activation transition, albeit less than S1 and S2, from ~17 to ~19 Å. Resampling statistics (bootstrapping[37]) were used to convert experimental variability into a confidence interval for the fit solutions. The DEPET-resolved distances in the human BK channel VSD are compared to homologous positions in the cryo-EM-resolved *Aplysia* BK channel in Supplementary Table 3.

Finally, the distances were combined to evaluate the orientation of the Trp side-chain with respect to the labeling site (Fig. 5a). The distances and orientation evaluated from DEPET data are completely agnostic of the protein structure. However, they are in very good agreement with those in the cryo-EM-determined structures of the BK channel voltage-sensing domain (Fig. 5b), in terms of the overall distance of the S4 Trp from

helices S0, S1, and S2, and the orientation of the S4 Trp side-chain, which faces towards S0 and away from S1 and S2. The finding is also consistent with the position of non-conserved helix S0 in the periphery of the BK channel VSD.

**The depolarization-evoked activation of the BK channel VSD.** The recent BK channel cryo-EM structures have provided invaluable insight on ligand-dependent channel activation and opening[20,21]. However, the voltage-sensing domains of unliganded/shut channels are virtually structurally identical to those of $Ca^{2+}$/$Mg^{2+}$-bound/open channels, and likely correspond to a domain in the resting conformation (Supplementary Figure 2). Implementing DEPET in conducting channels, expressed in cells

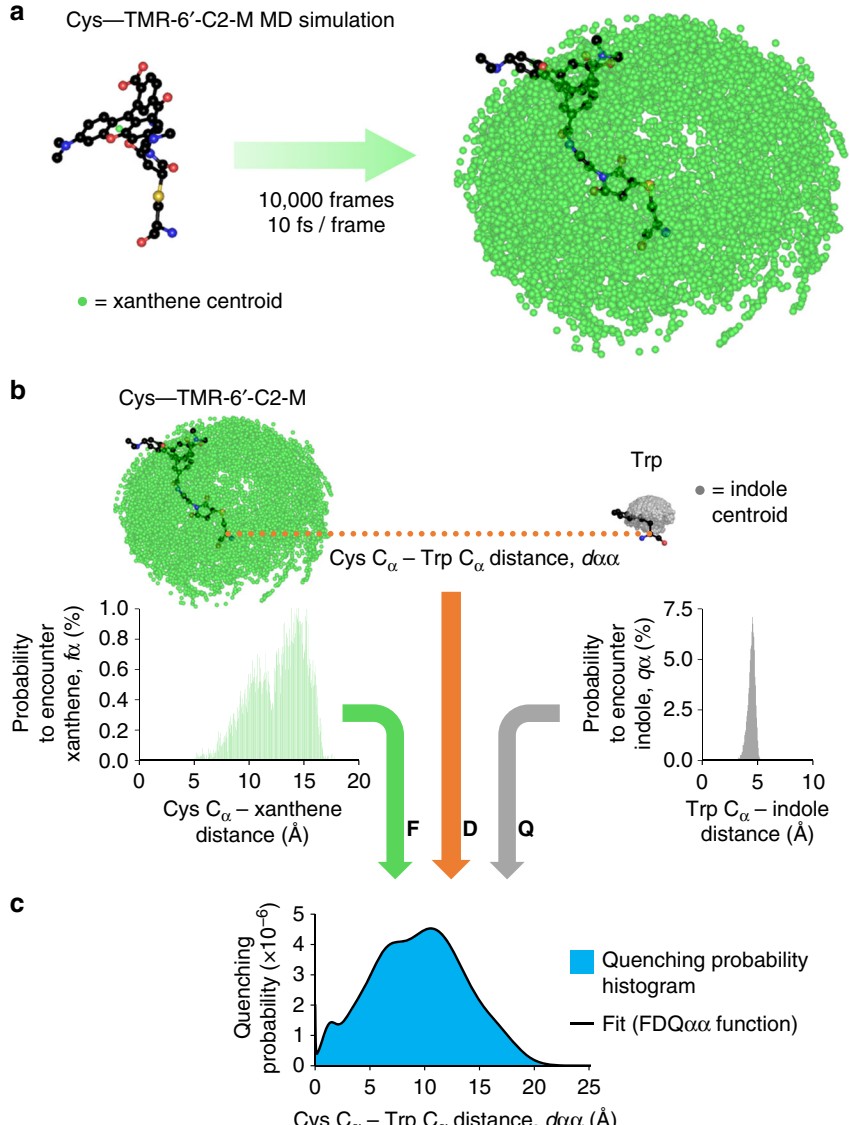

**Fig. 3** Producing distance-dependent quenching functions for DEPET. **a** A Cys—TMR-6′-C2-M conjugate was constructed and simulated using molecular dynamics (MD). The centroid of the fluorescent moiety (xanthene) is shown as a green sphere. **b** A histogram for the probability to encounter xanthene a given distance from the Cys $C_\alpha$, $f_\alpha$, was constructed by binning Cys $C_\alpha$/xanthene distances following the MD simulation. The same method was used to calculate the Trp side-chain (indole, gray) probability as a function of distance from the Trp $C_\alpha$, $q_\alpha$. **c** Since indole quenches xanthene by contact[8], the chance that they encounter each-other ($f_\alpha \cap q_\alpha$) is the quenching probability, which was calculated for any distance ($d_{\alpha\alpha}$) separating the $C_\alpha$ atoms of the labeled Cys and the Trp (blue). This histogram was empirically fit with a sum of exponential and Gaussian functions (black curve), subsequently used to extract distance information from DEPET data. The Fluorophore—Distance—Quencher ($FDQ_{\alpha\alpha}$) functions for the other TMRM fluorophores are shown in Supplementary Figure 7; the $FDQ_{\alpha\alpha}$ parameters are in Supplementary Table 1. A similar exercise to correlate quenching probability to the distance between the Cys $C_\alpha$ and Trp $C_\beta$ atoms ($FDQ_{\alpha\beta}$) is shown in Supplementary Figure 9; parameters in Supplementary Table 2

under voltage clamp and physiologically-relevant experimental conditions allowed the determination of the S4 voltage-dependent rearrangement relative to its surrounding helices (Fig. 5a). Can the atomic coordinates from cryo-EM be combined with the distance information from DEPET to resolve how the BK VSD activates? We imposed the DEPET-measured distance changes between Cys $C_\alpha$ and Trp $C_\alpha$ atoms as the VSD transitions from the resting to the activated state (i.e., $d_{\alpha\alpha,A} - d_{\alpha\alpha,R}$) to the structure of the unliganded/shut channel and asked: where does the S4 Trp go when the voltage sensor activates? Trilateration for the final coordinates of the S4 Trp yielded two sets of potential positions: 12.1 [95% CI 9.9, 14.7] Å above (along the z-axis of the structure) and 6.6 [4.6,8.9] Å below, into the membrane (Supplementary Figure 10). Since the movement of the positively-charged S4 helix

upon depolarization is expected to be outward[32–36,38], we favor the first set of solutions.

**Validating DEPET with length standards.** The agreement of DEPET data with those of the cryo-EM BK channel structure are highly encouraging. However, the BK structures are not the best standard to quantifiably evaluate the accuracy of DEPET, since their resolution was ~3.5 Å[20,21] and they resolved a molluscan BK channel with substantial divergence from the human protein[39]. As many scientific approaches, DEPET carries necessary, simplifying assumptions, which necessitates a more stringent determination of its accuracy, i.e., the goodness of the FDQ functions in extracting a distance measurement from fluorescence

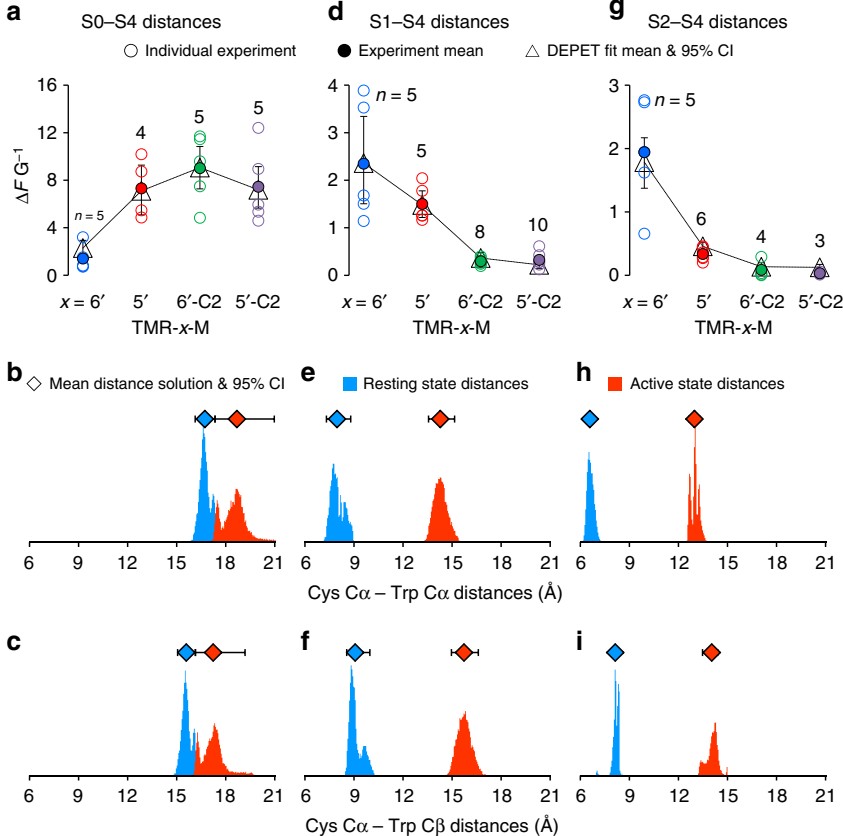

**Fig. 4** Distances in the BK channel before and after membrane depolarization. **a** DEPET data (conductance-normalized, voltage-dependent fluorescence change, $\Delta F/G$) from individual experiments (open circles) and mean (filled circles) for TMRM fluorophores of increasing length (left to right) conjugated to helix S0 (labeled Cys at position 19) and exhibiting state-dependent quenching by W203, at S4. Representative traces in Supplementary Figure 5. Open triangles represent mean DEPET fits; number of experimental replicates are shown next to the symbols. **b** Distributions for the DEPET solutions (10,000 boostrapped replicates) for the C19 (S0) and W203 (S4) $C_\alpha$ atom distance ($d_{\alpha\alpha}$) in the Resting (blue) and Active (red) conformation. The diamond symbols above the distributions represent their means. **c** The distance between the C19 $C_\alpha$ and W203 $C_\beta$ atoms ($d_{\alpha\beta}$), in the Resting (blue) and Active (red) conformation. $c$ (the coefficient to convert quenching probability to $\Delta F$ $G^{-1}$) was 1.38 [0.53,6.70] $\times 10^6$. Note that $d_{\alpha\alpha} > d_{\alpha\beta}$, suggesting that W203 points towards helix S0 in both Active and Resting conformations of the BK voltage-sensing domain. **d–f** DEPET data and fits, from TMRM fluorophores conjugated to position 136, extracellular to helix S1. Representative traces in Fig.2. $c = 3.57$ [2.34,5.00] $\times 10^5$. **g–i** DEPET data and fits, from TMRM fluorophores conjugated to position 145, extracellular to helix S2. Representative traces in Supplementary Figure 6. $c = 4.01$ [2.31,5.53] $\times 10^5$. Note that $d_{\alpha\alpha} < d_{\alpha\beta}$, for both S1 and S2, indicating that the S4 Trp side-chain points away from these helices (See Fig. 5). These distances are also shown compared to distances from the cryo-EM BK channel structures in Supplementary Table 3. A determination of the W203 $C_\alpha$ coordinates in the Active state of the BK channel VSD is shown in Supplementary Figure 10. All error bars are 95% CI

quenching. This is critical for the applicability of DEPET to a wide variety of biological molecules of unknown a priori structure. We used an approach previously implemented to validate FRET-based distance measurements: the use of rigid, polyproline-based peptides as length standards[3] (Fig. 6). Specifically, we sought to determine the length of synthetic peptides of the general formula Cys-(Pro)$_n$-Trp by (i) measuring intramolecular quenching of conjugated TMRM fluorophores of different length and (ii) correlating the observed quenching with a separating Cys/Trp distance using the $FDQ_{\alpha\alpha}$ functions, as performed on the BK channel. On average DEPET produced length estimates 1.3 Å off the expected length (Supplementary Table 4): that is, DEPET, in its current implementation using Trp as a collisional quencher and commercially-available TMRM fluorophores, can determine distances, and distance changes, with an exquisitely fine grain. If a different combination of fluorophore/quencher is used, it would be prudent to evaluate their quenching/distance response in the same way to ensure the goodness of the corresponding FDQ functions.

## Discussion

Combining DEPET distance constraints to a static cryo-EM-resolved structure of an ion channel demonstrates how two cutting-edge approaches can be combined to enhance our understanding of protein structure and function (Supplementary Figure 10). However, there are important caveats to consider while integrating the two approaches: DEPET measurements were performed in conducting, human BK channels, bearing mutation R207Q to enable the study of voltage-dependence, expressed in living cells; the cryo-EM studies were performed in purified molluscan BK channels that exhibit considerable sequence diversity from their mammalian homologs[39]. Nevertheless, a 12 Å vertical S4 translation is reasonable, considering that the archetypal Shaker K$^+$ channel S4 helix is thought to translate by ~15 Å[36]. This conformational reorganization represents the first response of the universal regulator of cellular excitability to membrane depolarization.

Importantly, the voltage-sensing properties of BK channels are modified in vivo by a multitude of biological factors, including

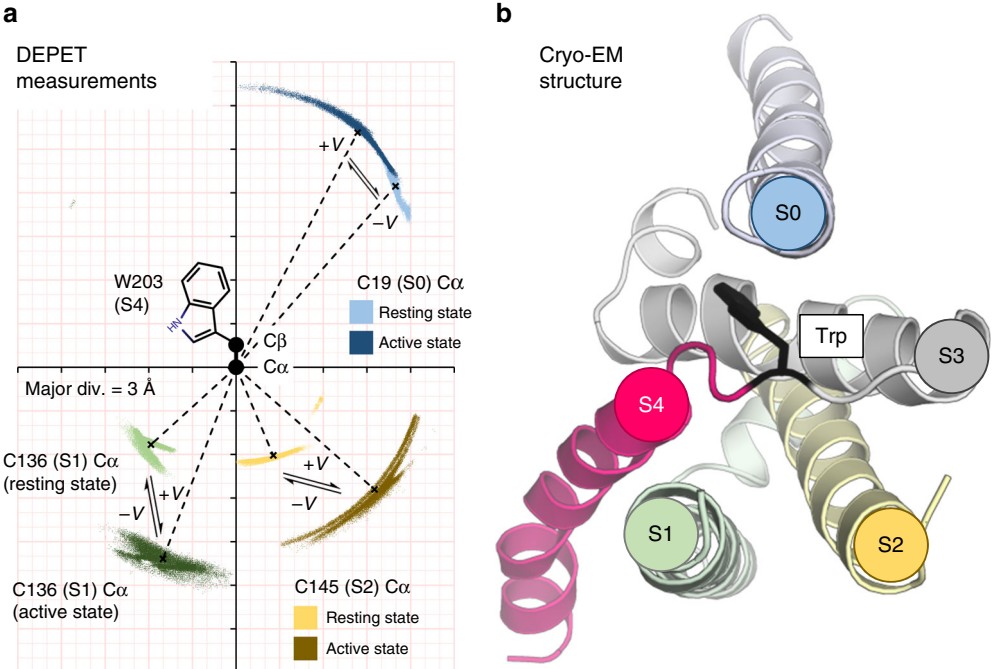

**Fig. 5** DEPET results in relation to the known structure of the BK VSD. **a** Distribution of the labeled Cys $C_\alpha$ atoms in the resting (position 19, S0: light blue; position 136, S1: light green; position 145, S2: yellow) and active (19: dark blue; 136: dark green; 145: dark yellow) conformations with respect to the $^{203}$Trp $C_\alpha$ and $C_\beta$ atoms, calculated by the DEPET distance measurements (Fig. 4). The dashed lines indicate the mean $d_{\alpha\alpha}$ distances in the resting and active states (Fig. 4b, e, h). The angle formed by the $^{19}$Cys $C_\alpha$–$^{203}$Trp $C_\alpha$–$^{203}$Trp $C_\beta$ atoms was 41° [37°,48°] in the resting conformation and 28° [10°,38°] in the active conformation. The orientation of $^{203}$Trp side-chain with respect to $^{136}$Cys in S1 was determined to be 132° [123°,142°] in the resting conformation and 159° [148°,172°] in the active conformation. Finally, the orientation of $^{203}$Trp with respect to $^{145}$Cys in S2 was 157° [115°,175°] in the Resting state and 132° [108°,157°] in the Active state. That is, the side-chain of W203 (helix S4) points towards helix S0 and away from helices S1 and S2, and its distance increases upon voltage-dependent activation. **b** Top view of an isolated voltage-sensing domain from the cryo-EM-derived structure of the *Aplysia* ligand-free/shut BK channel (PDB: #5TJI[21]). W192 (homologous to W203 in the human channel[20]) is shown with its side-chain, in black

allosteric contributions from intracellular ligands such as $Ca^{2+}$, $Mg^{2+}$, and heme; extracellular cofactors such as $Cu^{2+}$, and the association of auxiliary β and γ subunits (reviewed in refs. [18,19]). We postulate that this functional modification of BK channel properties has a structural basis, underpinned by remodeling of the voltage-sensing domain and its activation transition. DEPET, an approach that provides in cellula structural information under flexible and physiological conditions is an ideal tool to investigate how the BK response to membrane depolarization is modulated, to finely tune its activity in a broad spectrum of signaling millieux.

While our first protein implementation of DEPET was on membrane-bound ion channels, the successful measurement of peptide lengths means that it is generalizable to a wide spectrum of structural biology problems. DEPET possesses important advantages compared to other optical approaches: (i) It offers the capability to measure distances from 0 nm (in theory; 0.6 nm in this work) to ~2.5 nm with a particularly fine grain. While we were able to achieve this using commercially-available TMRM labels, the use of shorter or longer fluorophores with a more contoured FDQ function may yet improve on the accuracy and range of DEPET (but see limitation, below). (ii) DEPET provides a direct measurement of distances, and distance changes, between protein backbone ($C_\alpha$) and side-chain ($C_\beta$) atoms, instead of fluorescence donor and acceptor moieties. This obviates the need for post hoc homology and all-atom protein modeling, necessary to translate FRET-determined distances to measurements relevant to the protein structure. Finally, (iii) DEPET allows the real-time measurement of protein function and structure, under

physiologically-relevant and flexible experimental conditions, without necessitating large fluorescent protein adjuncts or millimolar concentrations of transition or lanthanide metal acceptors.

DEPET also has limitations that should be considered: (i) it is susceptible to steric interference between the fluorophore and Trp. This is more likely to occur when measuring longer distances, in which case FRET-based approaches are preferable. (ii) Cell autofluorescence and non-specific labeling can influence the measurement of background fluorescence in membrane-embedded proteins. This is why, in the BK channel implementation, we used the fluorescence change following the structural rearrangement of the protein: a signal that only arises from protein-conjugated fluorophores (Supplementary Figure 4c). The latter is not a problem in purified labeled proteins (as in the case of polyproline peptides, Fig. 6). In addition, special measures can be taken to minimize the extent of non-specific background fluorescence in membrane-expressed proteins[40].

A challenge in the implementation of DEPET is the fluorescent labeling of accessible protein loci. For instance, in membrane-bound proteins only the extracellular portions are available for fluorescence labeling. The exciting advent of patch clamp fluorometry[40,41] and the increasing availability of fluorescent unnatural amino acids (fUAA)[42–44] enable the labeling of membrane-bound proteins at any position. In a new protein, especially one of unknown structure, finding appropriate positions for fluorescence labeling (Cys/fUAA) and quenching (Trp) that result in DEPET signals without interfering with protein function can be a laborious game of molecular Battleship. However, once a pair of fluorescence labeling/quenching positions has

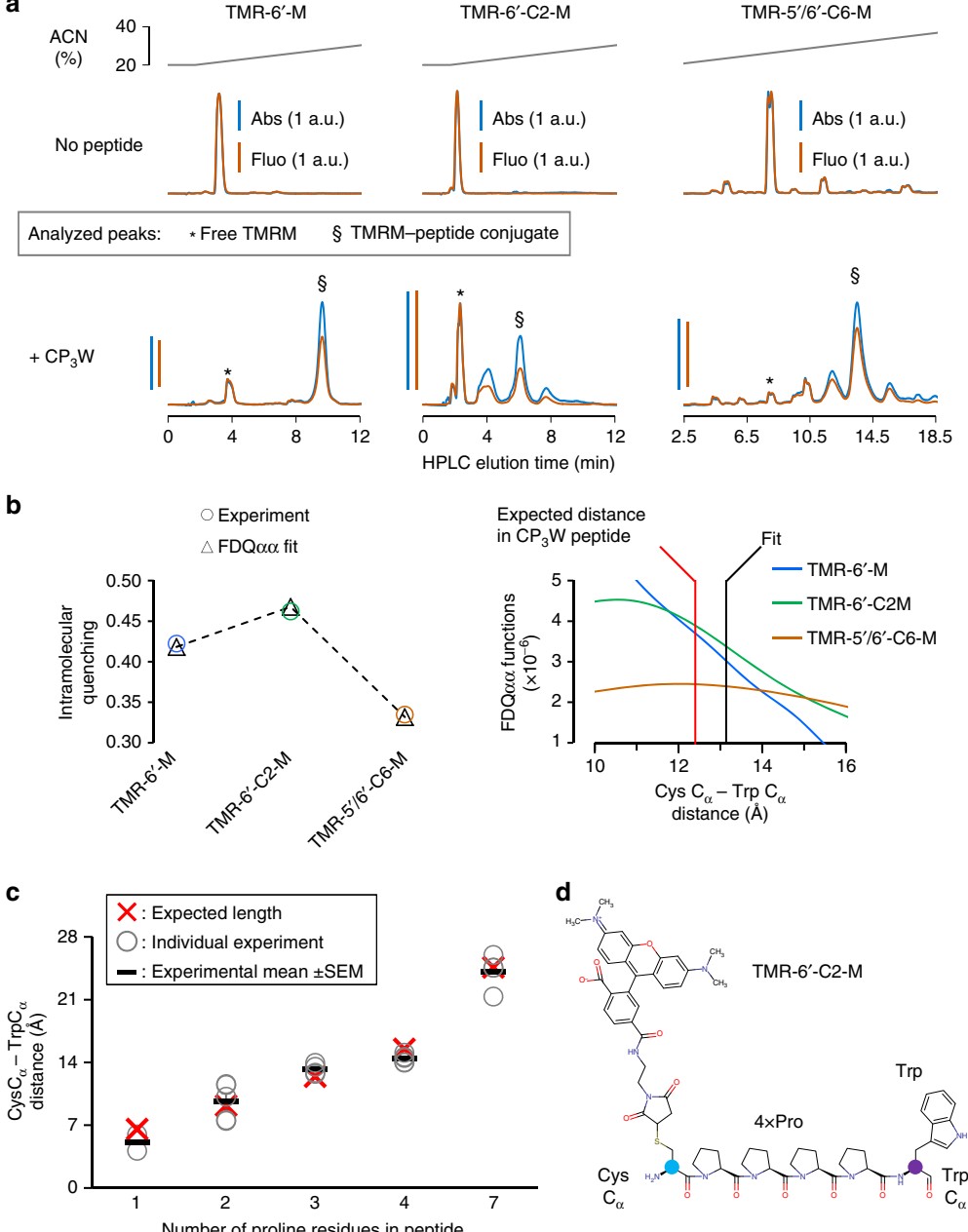

**Fig. 6** Evaluating DEPET accuracy using peptide length standards. **a** Reversed-phase HPLC chromatograms of free TMRM fluorophore and fluorophore/polyproline peptide conjugates. Gray: acetonitrile (ACN) content of the elution buffer; blue: TMRM absorbance ($\lambda = 550$ nm); orange: TMRM fluorescence ($\lambda_{ex} = 550$ nm; $\lambda_{em} = 575$ nm). Note that the presence of a Cys-Pro$_3$-Trp peptide (i) depletes the free TMRM fluorophore; (ii) gives rise to conjugate species, shown as new elution bands; and (iii) the conjugate species exhibit less fluorescence, for the same amount of absorbance; i.e., they are quenched. The most abundant conjugate species were analyzed further, marked by asterisks. **b** Intramolecular quenching of TMRM fluorophores of increasing length conjugated to the CP$_3$W peptide (circles). The fits of these data to the FDQ$_{\alpha\alpha}$ functions (see Fig. 3c & Supplementary Figure 7c) are shown as triangles. The Cys C$_\alpha$–Trp C$_\alpha$ distance was estimated to be 13.1 Å; in this peptide with three prolines, it was expected to be 12.4 Å. The c parameter was $1.38 \times 10^5$. The expected and fit distances in relation to the FDQ functions are shown on the right. **c** Summary of all peptide length determination experiments. The number of experimental replicates were 2, 5, 4, 5, 4 for peptides with 2, 3, 4, 5 or 7 Pro, respectively. Open circle: individual experiment; dash: experimental mean; red cross: nominal length. On average, the FDQ-determined length estimates were off by ~1.3 Å (see Supplementary Table 4). **d** Molecular structure of a Cys-Pro$_4$-Trp peptide conjugated to the TMR-6′-C2-M fluorophore. The Cys and Trp C$_\alpha$ atoms are indicated in blue and purple, respectively

been identified, DEPET provides model-independent and accurate intramolecular distances, distance changes and side-chain orientations, directly associable with protein function. We believe that DEPET is a valuable addition to the armory of the structural biologist or molecular physiologist, either stand-alone or in combination with complementary approaches.

## Methods

**Determining Trp quenching efficiency for TMRM fluorophores.** Trp residues efficiently quench the fluorescence of rhodamine-based fluorophores using a well-characterized $e^-$ exchange mechanism: PET[8,9]. In distance-encoding PET (DEPET), distance information is extracted from the differential state-dependent quenching of fluorophores of different length, when they are conjugated on the same labeling site (a substituted Cys), by a nearby Trp residue (Fig. 1). It is therefore important to

ascertain whether Trp is equally efficient at quenching all fluorophores used in the study. In this work, we used thiol-reactive fluorophores of the tetramethylrhodamine-maleimide (TMRM) type, which are commercially available at different lengths. From shortest to longest: TMR-6′-M (Anaspec); TMR-5′-M (Anaspec); TMR-6′-C2-M (Anaspec); TMR-5′-C2-M (Anaspec); TMR-6′-C6-M and TMR-5′-C6-M (Biotium). Note that the last two were only available as mixed isomers, and are referred to as TMR-5′/6′-C6-M.

Fluorophore stocks were dissolved in anhydrous DMSO (Thermo Fisher Scientific) to 100 mM and kept at −20 °C in a dessicator. Solutions were made including each TMRM fluorophore (0.5 μM) and final [Trp] (0, 2.5, 5, 10, 15, 20, 25, and 30 μM), in voltage clamp extracellular solution (see Voltage clamp fluorescence spectroscopy section, below). [Trp] was determined by its absorbance at 280 nm, using extinction coefficient $\varepsilon = 5500$ M$^{-1}$ cm$^{-1}$. Fifty microliters aliquots were added in quadruple, over two 96-well plates suitable for fluorescence measurements (Corning; black polystyrene, flat, clear bottom). Absorbance (260–600 nm) and fluorescence ($\lambda_{ex} = 540$ nm; $\lambda_{em} = 565$–700 nm) were measured for each well in a Synergy H1 plate reader (BioTek Instruments) (Supplementary Figure 1). Stern-Volmer plots[1] were constructed for each fluorophore: $F_0/F$ (where $F$ is fluorescence at 575 nm, normalized by absorbance at 550 nm; $F_0$ is $F$ in Trp-free solution) plotted against [Trp]. Linear regression was performed in MATLAB (MathWorks). All fluorophores had Stern-Volmer bimolecular quenching constants ($K_{SV}$) in the range of ~36–42 M$^{-1}$; as such, they are considered equally quenched by Trp. We suggest that, when fluorophores of different chemistry are used, which may therefore exhibit significantly different Trp-induced quenching efficiency, their $K_{SV}$ can be used to scale the quenching probability distribution (FDQ) functions (see below).

**Intra-domain BK channel Ca$^{2+}$- and Mg$^{2+}$-induced transitions**. The BK channel cryo-EM-resolved structures from *Aplysia californica* in the presence[20] or absence[21] of Ca$^{2+}$ and Mg$^{2+}$ ligands (PDB: #5TJ6 and #5TJI, respectively) were loaded on PyMOL (Schrödinger). Residues not shared by both structures were excluded. Each functional domain (VSD: residues 15–214; pore domain: 215–316; cytosolic: 317–1065) were isolated and aligned using the PyMOL align function. Pairwise C$_\alpha$-C$_\alpha$ distances for each residue in each domain were measured for the apo and ligand-bound states. A color-coded cartoon of the structure and the per-residue relative movements are shown in Supplementary Figure 2.

**Voltage clamp fluorescence spectroscopy**. Molecular biology: A human BK channel (hSlo) clone (#U11058)[45] transcribed from the fourth Met without extracellular Cys (C14S, C141S, C277S) was used. Background mutation R207Q[30] was introduced to increase $P_O$ at low [Ca$^{2+}$]$_i$ and allow full characterization of the voltage dependence. A single Cys was substituted at the extracellular flank of S0 (Q19C), S1 (N136C), or S2 (Y145C) for subsequent modification by thiol-reactive fluorescent labels. In control experiments, the native tryptophan at the S3–S4 extracellular flank was substituted by valine (W203V) to ascertain that the resolved $\Delta F$ is due to the state-dependent interaction of the conjugated dye with W203. To test against inter-subunit quenching, BK subunits without extracellular Cys were coexpressed with subunits including an introduced Cys (N136C), without W203 (W203V). Mutations were generated with QuikChange Site-Directed Mutagenesis Kit (Agilent) and confirmed by sequencing. cDNA was transcribed to cRNA in vitro (mMESSAGE MACHINE, Thermo Fisher Scientific) and stored at −80 °C in RNA storage solution (Thermo Fisher Scientific).

Oocyte preparation: *Xenopus laevis* (Nasco) oocytes (stages V–VI) were ethically isolated and defolliculated using standard procedures[46]. UCLA's animal care and use program has been fully accredited by the Association for Assessment and Accreditation of Laboratory Animal Care International continuously since 1976. The oocytes were injected with 50 nl of cRNA encoding for the human Slo1 (BK) channel clones described above (0.1–0.5 ng/nl) using a Drummond nanoinjector. Injected oocytes were maintained at 18 °C in an amphibian saline solution supplemented with 100 units/ml penicillin, 100 μg/ml streptomycin and 50 μg/ml gentamicin (Thermo Fisher Scientific). Twenty-four hours before experimenting (2–5 days after injection), DTT (200 μM) and EDTA (10 μM) were added to the oocyte solution, to make Cys available for fluorophore labeling. On the day of experiments, oocytes were rinsed in DTT- and EDTA- free saline and stained for 60 min with 2 μM TMRM fluorophores in a depolarizing solution (in mM: 120 K-methanesulfonate (MeS), 2 Ca(MeS)$_2$, and 10 HEPES, pH = 7.0) on ice, in the dark, to label the introduced Cys. The oocytes were then rinsed in dye-free saline prior to being mounted in the recording chamber.

Electrophysiology: Cut-open oocyte Vaseline gap (COVG) is a low-noise, fast voltage clamp technique[23,24]. The oocyte is placed in a triple-compartment Perspex chamber, with a diameter of 600 μm for the top and bottom apertures. The upper chamber isolates the oocyte's upper domus and maintains it under clamp. The middle chamber provides a guard shield by clamping the middle part of the oocyte to the same potential as the upper chamber. The bottom chamber injects current intracellularly, through the saponin-permeabilized part of the oocyte. Fluorescence emission and ionic current are simultaneously measured from the same area of membrane isolated by the top chamber[24,25]. The optical setup consists of a Zeiss Axioscope FS microscope with filters (Semrock Brightline) appropriate for rhodamine excitation and emission wavelengths. The light source is a 530 nm, 158 lm Luxeon Rebel LED. A TTL-triggered Uniblitz VS 25 shutter (Vincent

Associates) is mounted on the excitation lightpath. The objective (Olympus LUMPlanFl, 40×, water immersion) has a numerical aperture of 0.8 and a working distance of 3.3 mm, which leaves enough room for the insertion of the microelectrode while fully covering the oocyte domus exposed in the external recording chamber. The emission light is focused on a PIN-08-GL photodiode (UDT Technologies). A Dagan Photomax 200 amplifier is used for the amplification of the photocurrent and background fluorescence subtraction. External solution (mM): 120 Na-MeS, 10 K-MeS, 2 Ca(MeS)$_2$, 10 HEPES (pH = 7.0). Internal solution (mM): 120 K-glutamate, 10 HEPES (pH = 7.0). Intracellular micro-pipette solution (mM): 2700 Na-MeS, 10 NaCl. Low access resistance to the oocyte interior was obtained by permeabilizing the oocyte with 0.1% saponin carried by the internal solution. To limit experimental variation, all experiments analyzed for any given labeling position (≥3 per fluorophore) were performed on the same batch of oocytes, i.e., collected at the same time, from the same frog.

The oocyte membrane holding potential was −90 mV. The membrane potential was pulsed from −220 to +160 mV for 50 ms in 20 mV increments with four averaging pulses per test potential. Test pulses were flanked by pre-pulses and post-pulses to −160 mV (300 and 100 ms, respectively). Pulse cycle period was 2 s.

Initial analysis: The procedure is demonstrated in Supplementary Fig. 3. Fluorescence bleaching was excluded by subtracting an exponential function fit to a recording without a voltage pulse. The total voltage-dependent fluorescence change ($\Delta F_{total}$) was calculated by fitting the amplitude of voltage-evoked fluorescence deflections ($\Delta F$) with a Boltzmann function, by least squares, in Excel:

$$\Delta F = \frac{\Delta F_{total}}{1 + \exp\left[\frac{zF}{RT}(V_{0.5} - V_m)\right]} + \Delta F_{min} \qquad (1)$$

where $V_m$ is the membrane potential; $V_{0.5}$ is the half-activation potential; $z$ is the effective valence; $F$ and $R$ the Faraday and Gas constants, respectively; $T$ is temperature (294 K).

The $\Delta F_{total}$ was normalized for channel expression using the maximal macroscopic conductance, $G_{max}$. The latter was extracted by fitting the macroscopic conductance with the sum of two Boltzmann functions:

$$G = \sum_{i=1}^{2} \frac{G_{max,i}}{1 + \exp\left[\frac{z_iF}{RT}(V_{0.5,i} - V_m)\right]} \qquad (2)$$

Macroscopic conductance, $G$, was calculated by dividing the current ($I$) by the driving force:

$$G = \frac{I}{V_m - E_K} \qquad (3)$$

where $E_K$ is the equilibrium potential for potassium. Finally,

$$G_{max} = \sum_{i=1}^{2} G_{max,i} \qquad (4)$$

**Constructing the FDQ quenching probability functions**. DEPET is uniquely capable of estimating distances between assigned atoms of the protein backbone or side-chains. In this implementation, we describe how the distances between (i) the fluorescently-labeled Cys C$_\alpha$ atom and the Trp C$_\alpha$ atom, which is an intramolecular distance between atoms of the protein backbone; and (ii) the Cys C$_\alpha$ atom and the Trp C$_\beta$ atom; to inform on the orientation of the Trp side-chain with respect to the label site, and extract relative atomic coordinates. For brevity, we refer to the Cys C$_\alpha$ atom as Cα, the Trp C$_\alpha$ atom as Wα and the Trp C$_\beta$ atom as Wβ.

For a PET quencher such as Trp, the probability to contact the fluorophore is effectively the quenching probability ($P_Q$)[8,13]. In DEPET, the optical signals correspond to a change in Trp-induced $P_Q$. We seek to construct a function that will correlate the change in $P_Q$ to a change in distance, between two positions along the protein backbone: Cα and Wα atoms.

How likely is it to encounter (and therefore, quench) the fluorophore at a given distance from the Cα? First, we need to characterize fluorophore range and flexibility. This is achieved by molecular dynamics (MD) simulations. A Cys-TMRM conjugate was designed in MarvinSketch (ChemAxon) (Fig. 3a). The Cys amino and carboxy groups were neutralized to simulate a non-terminal residue. The conjugate's molecular structure was energy-minimized and then underwent a MD simulation in a desktop PC running MarvinSketch, using the MMFF94 forcefield[47] with Velocity Verlet integrator and initial temperature 300 K. Total simulation time was 100 ps with frames saved every 10 fs, collecting 10,000 frames per fluorophore (Fig. 3a). The position of the centroid of the fluorescent moiety (xanthene, in the case of the TMRM fluorophores) was extracted, from each frame, in PyMOL, by averaging the Cartesian coordinates of each constituent atom. Finally, the distances between the Cα and the fluorophore centroid were measured in each frame, also in PyMOL, and collected into a histogram with a bin size of 0.05 Å. Each bin was divided by the total number of observations (10,000) to produce the probability distribution $f_\alpha(X,K)$, reflecting the frequency to encounter

fluorophore $X$ within volume interval $K$, corresponding to a spherical shell with its center at the C$\alpha$, maximal radius $R$ and thickness $\delta R = 0.05$ Å (Fig. 3b, Supplementary Figure 7b).

The Trp indole side-chain, i.e., the quenching moiety, is also expected to undergo some thermal isomerization. The above procedure was repeated for Trp, constructing a histogram of the distance of the indole centroid from the W$\alpha$. This probability distribution is defined as $q_\alpha(\kappa)$, where $q_\alpha$ is the probability of Trp side-chain encounter within spherical shell $\kappa$, which has a center at the W$\alpha$ atom, maximal radius $r$ and thickness $\delta r = 0.05$ Å (Fig. 3b).

The quenching probability of fluorophore $X$ over distance $d_{\alpha\alpha}$ separating the Cys and Trp C$\alpha$ atoms, $P(X, d_{\alpha\alpha})$ corresponds to the intersection of $f_\alpha$ and $q_\alpha$, scaled by their available volume of interaction for each interval of separating distance (Fig. 3c, Supplementary Figure 7c):

$$P(X, d_{\alpha\alpha}) = \sum_{R,r} \frac{V_{K \cap \kappa}(d_{\alpha\alpha}, R, r)}{V_K(R)} f_\alpha(X, K) \times \frac{V_{K \cap \kappa}(d_{\alpha\alpha}, R, r)}{V_\kappa(r)} q_\alpha(\kappa) \quad (5)$$

$V_K(R)$ and $V_\kappa(r)$ are the volumes of spherical shells with maximal radii $R$ and $r$, respectively, and thickness $\delta r = 0.05$ Å. Their general formula is:

$$V(\rho) = \frac{4\pi}{3} \left[ \rho^3 - (\rho - \delta r)^3 \right] \quad (6)$$

$V_{K \cap \kappa}(d, R, r)$ is the intersection volume of shells K and $\kappa$, each with maximal radii $R$ and $r$, respectively, thickness $\delta r = 0.05$ Å, and their centers separated by distance $d$. Its calculation depends on the values of the $d$, $R$ and $r$ variables:

If $d = 0$ and $R \neq r$; or $d \geq R + r$; or $d \leq R - r - \delta r$; or $d \leq r - R - \delta r$:

$$V_{K \cap \kappa}(d, R, r) = 0 \quad (7)$$

If $d = 0$ and $R = r$:

$$V_{K \cap \kappa}(d, R, r) = V_K(R) = V_\kappa(r) \quad (8)$$

If $d = R + r - \delta r$:

$$V_{K \cap \kappa}(d, R, r) = V_\varphi(d, R, r) \quad (9)$$

where $V_\varphi(d, R, r)$ is the volume of the lens formed by the intersection of two spheres with radii $R$ and $r$, whose centers are separated by distance $d$. Its general formula is:

$$V_\varphi(\chi, \psi, \omega) = \frac{\pi(\psi + \omega - \chi)^2(\chi^2 + 2\chi\psi - 3\psi^2 + 2\chi\omega - 3\omega^2 + 6\psi\omega)}{12\chi} \quad (10)$$

where $\chi$ is the distance separating the centers of two spheres with radii $\psi$ and $\omega$.
If $d = R - r$:

$$V_{K \cap \kappa}(d, R, r) = V_\varphi(d, R, r) - V_\varphi(d, R - \delta r, r) \quad (11)$$

If $d = r - R$:

$$V_{K \cap \kappa}(d, R, r) = V_\varphi(d, R, r) - V_\varphi(d, R, r - \delta r) \quad (12)$$

Else, in all other cases:

$$V_{K \cap \kappa}(d, R, r) = V_\varphi(d, R, r) - V_\varphi(d, R - \delta r, r) - V_\varphi(d, R, r - \delta r) + V_\varphi(d, R - \delta r, r - \delta r) \quad (13)$$

Note that the two longest fluorophores, TMR-5′-C6-M and TMR-6′-C6-M, were only available as mixed isomers (Biotium). The $P(X, d_{\alpha\alpha})$ for each was calculated separately, then they were summed and divided by 2, to construct their joint distribution (Supplementary Figure 7c, orange).

The result of the above calculations is the quenching probability for each Cys-fluorophore conjugate by a Trp when their respective C$\alpha$ separated over distance $d_{\alpha\alpha}$, discretized in 0.05-Å bins. In order for this probability distribution to be usable by curve-fitting algorithms, it was empirically fit by the sum of up to two exponential and six Gaussian functions:

$$\mathrm{FDQ}_{\alpha\alpha}(X, d_{\alpha\alpha}) = \sum_{i=1}^{2} \alpha_i \exp\left(-\frac{d}{\delta_i}\right) + \sum_{j=1}^{6} \frac{A_j}{\sqrt{2\pi}\sigma_j} \exp\left[-\frac{(d - \mu_j)^2}{2\sigma_j^2}\right] \quad (14)$$

where $\alpha$ and $A$ are amplitude factors for the exponential and Gaussian distribution functions, respectively; $\delta$ is the length constant of the exponential function; and $\mu$ and $\sigma$ are the mean and standard deviation of the Gaussian distribution function, respectively. Parameters for each fluorophore are reported in Supplementary

Table 1. The histograms, and their fits, are shown in Fig. 3c and Supplementary Figure 7c. We refer the above functions as Fluorescence—Distance—Quencher (FDQ) functions.

In this work, we use photochemically-identical TMRM fluorophores, which exhibit nearly identical PET-mediated bimolecular quenching by Trp (Supplementary Figure 1). Note that, if other fluorophores are used, their FDQ functions can be scaled by their $K_{SV}$, to eliminate the differential quenching efficiency variable from the distance measurement.

In addition to the C$\alpha$W$\alpha$ (or $d_{\alpha\alpha}$) distance, DEPET optical data can be used to evaluate other distances pertinent to protein structure: for instance the distance between the labeled Cys C$\alpha$ (C$\alpha$) and the Trp C$\beta$ atom—the first atom of the Trp side-chain (W$\beta$). This information is useful to evaluate the orientation of the Trp side-chain with respect to the labeled Cys: if the Trp points towards the labeled Cys, then C$\alpha$W$\beta$ < C$\alpha$W$\alpha$, and vice versa.

To determine C$\alpha$W$\beta$ (or $d_{\alpha\beta}$), new FDQ functions are required. We begin by evaluating the probability to encounter the Trp side-chain (indole) centroid a given distance from W$\beta$ ($q_\beta$; Supplementary Figure 9a). As expected, this probability density is distributed over a shorter distance than $q_\alpha$ (Supplementary Figure 9a). We repeated the exercise of constructing $\mathrm{FDQ}_{\alpha\alpha}(X, d_{\alpha\alpha})$ (Eqs. (5–13)), now using the $q_\beta$ distribution instead of $q_\alpha$, for each fluorophore ($X$). Accordingly, we constructed distributions $\mathrm{FDQ}_{\alpha\beta}(X, d_{\alpha\beta})$, which denote the probability to quench fluorophore $X$ attached to C$\alpha$ as a function of distance $d_{\alpha\beta}$, or C$\alpha$W$\beta$ (Supplementary Figure 9b-c). $\mathrm{FDQ}_{\alpha\beta}$ distributions were also fit to sums to exponential and Gaussian functions (Eq. (14)); their parameters are listed in Supplementary Table 2.

Using a similar strategy, it is possible to construct $\mathrm{FDQ}_{\beta\alpha}$ and $\mathrm{FDQ}_{\beta\beta}$ distributions, to correlate PET fluorescence quenching as a function of distances C$\beta$W$\alpha$ and C$\beta$W$\beta$, respectively.

**Extracting distances and orientations from DEPET data.** The FDQ functions correlate quenching probability to a distance separating Cys and Trp atoms. In the BK channel experiments, the change in membrane potential resulted in fluorescence deflections ($\Delta F$), reflecting a change in Trp-induced quenching; so we are asking the question: what are the distances between the labeled Cys C$\alpha$ (position 136, helix S1) and the Trp C$\alpha$ (position 203, helix S4) in the Resting and Active conformations of the BK voltage-sensing domain? The following section uses nomenclature pertinent to conformational changes in the voltage-sensing domain; however, the same principle can be applied to determine intermolecular distances in any protein undergoing a conformational rearrangement surveyed by DEPET.

Consider a voltage sensor domain transitioning, upon membrane depolarization, from the Resting to the Active conformation. Fluorophore $X$ has been conjugated to a strategic position such that, upon depolarization, a change in fluorescence is observed ($\Delta F$) that is dependent on the presence of a nearby Trp residue:

$$\Delta F_X = F_A - F_R \quad (15)$$

This change in fluorescence is proportional to a change in Trp-induced quenching ($q_R$ and $q_A$, for quenching in the Resting and Active states, respectively) at the microscopic level:

$$\Delta F_X \propto (1 - q_A) - (1 - q_R) = (q_R - q_A)c \quad (16)$$

where $c$ is a coefficient to convert change in quenching probability to conductance-normalized $\Delta F$ data. Change in Trp quenching probability can be expressed as a function of the C$\alpha$W$\alpha$ ($d_{\alpha\alpha}$) distance:

$$\Delta F_X = \left[ \mathrm{FDQ}_{\alpha\alpha}\left(X, d_{\alpha\alpha,\mathrm{R}}\right) - \mathrm{FDQ}_{\alpha\alpha}\left(X, d_{\alpha\alpha,\mathrm{A}}\right) \right] c \quad (17)$$

where $\mathrm{FDQ}_{\alpha\alpha}(X, d_{\alpha\alpha,\mathrm{R}})$ and $\mathrm{FDQ}(X, d_{\alpha\alpha,\mathrm{A}})$ are the probabilities of fluorophore $X$ to be quenched by Trp, when the labeled Cys and Trp C$\alpha$ atoms are separated by distance $d_{\alpha\alpha,\mathrm{R}}$ (in the Resting state) or $d_{\alpha\alpha,\mathrm{A}}$ (in the Active state), respectively. That is, given a $\Delta F$ datum from fluorophore $X$, and its FDQ function, it is possible to extract the distance between the fluorophore and the Trp in the Resting and Active state ($d_R$ and $d_A$, respectively)—as well as coefficient $c$, which converts probability to conductance-normalized fluorescence units (and therefore carries no functional information). By simultaneously fitting the $\Delta F$ of multiple fluorophores, the solutions for $d_R$, $d_A$, and $c$ are greatly constrained.

To convert experimental variability of $\Delta F$ into a confidence interval for $d_R$ and $d_A$, bootstrapping (i.e., random sampling with replacement[37]) was used. The $G_{max}$-normalized $\Delta F_{total}$ data are bootstrapped to yield 10,000 sample sets (MATLAB Statistics and Machine Learning Toolbox), each including a bootstrap-averaged $\Delta F$ from every fluorophore used. This dataset is fit to:

$$\Delta F_{X,i} = \left[ \mathrm{FDQ}_{\alpha\alpha}\left(X, d_{\alpha\alpha,\mathrm{R},i}\right) - \mathrm{FDQ}_{\alpha\alpha}\left(X, d_{\alpha\alpha,\mathrm{A},i}\right) \right] c_i \quad (18)$$

where the $X$ refers to each fluorophore used, and $i = 1..10{,}000$, representing each of the 10,000 bootstrap sample sets. Fitting is performed using MATLAB's non-linear least squares solver (lsqcurvefit; Optimization Toolbox).

To prevent convergence to local error minima, the initial guess for each free parameter is seeded in five increments. Fitting one distance (CαWα, i.e., $d_{\alpha\alpha}$) results in a combined $5^3 = 125$ initial guess seeds for the $d_{\alpha\alpha,R}$, $d_{\alpha\alpha,A}$, and $c$ free parameters (Eq. 18). All 125 initial guess seeds are used to fit each of the 10,000 bootstrap sample sets; for each set, only the fit with the least error is saved, while the rest are discarded. Thus, the fitting routine performs 1,250,000 fits, yielding 10,000 solutions for each of $d_{\alpha\alpha,R}$, $d_{\alpha\alpha,A}$, and $c$.

If more distances are fit simultaneously, the following system of equations is used:

$$\Delta F_{X,i} = \left[ \text{FDQ}_{\alpha\alpha}\left(X, d_{\alpha\alpha,R,i}\right) - \text{FDQ}_{\alpha\alpha}\left(X, d_{\alpha\alpha,A,i}\right) \right] c_i$$
$$\Delta F_{X,i} = \left[ \text{FDQ}_{\alpha\beta}\left(X, d_{\alpha\beta,R,i}\right) - \text{FDQ}_{\alpha\beta}\left(X, d_{\alpha\beta,A,i}\right) \right] c_i \quad (19)$$

This fitting routine has five free parameters: the CαWα distances in the Resting and Active conformation ($d_{\alpha\alpha,R}$, $d_{\alpha\alpha,A}$); the CαWβ distances in the Resting and Active conformation ($d_{\alpha\beta,R}$, $d_{\alpha\beta,A}$); and $c$, which has the same value both FDQ sets. Each parameter is fit independently, but distance solutions that do not comply to atomic constraints, i.e., the difference between the $d_{\alpha\alpha}$ and $d_{\alpha\beta}$ distances should be less than the length of a single C–C bond (i.e., the distance between Wα and Wβ, 1.54 Å), are excluded.

Using the fit CαWα and CαWβ distances, it is possible to quantify the orientation of the Trp side-chain with respect to the labeled Cys $C_\alpha$ atom, (i.e., the $C\alpha\hat{W}\alpha W\beta$ angle), determined by the law of cosines:

$$C\alpha\hat{W}\alpha W\beta = \cos^{-1}\left( \frac{CC^2 + C\alpha W\alpha^2 - C\alpha W\beta^2}{2 \cdot CC \cdot C\alpha W\alpha} \right) \quad (20)$$

where CC is the length of a single C–C bond (1.54 Å), in this case representing the WαWβ distance in the CαWαWβ triangle. Likewise, we can determine the coordinates of Cα on a Cartesian plane, where Wα is at the origin (0,0) and Wβ at (0,1.54):

$$x_{C\alpha} = C\alpha W\alpha \cdot \cos\left(\tfrac{\pi}{2} - C\alpha\hat{W}\alpha W\beta\right)$$
$$y_{C\alpha} = C\alpha W\alpha \cdot \sin\left(\tfrac{\pi}{2} - C\alpha\hat{W}\alpha W\beta\right) \quad (21)$$

**Combining cryo-EM and DEPET information in the BK VSD**. Starting with the BK VSD structure captured in ligand-free, closed channels by cryo-EM[21], we sought to predict the position of the S4 Trp $C_\alpha$ atom when the VSD has been activated by membrane depolarization. However, the positions homologous to those labeled in this study were not all resolved in the structure, and the distances among nearby positions do not precisely agree (Supplementary Table 3). Therefore, the DEPET-measured distance changes between Cys $C_\alpha$ and Trp $C_\alpha$ atoms were used as the VSD transitions from the resting to the activated state (i.e., $d_{\alpha\alpha,A} - d_{\alpha\alpha,R}$; δ values in Supplementary Table 3a). That is, we used trilateration to answer the question: Given (i) the coordinates of S4 Trp $C_\alpha$ atom, and the $C_\alpha$ atoms of labeled positions in helices S0, S1, and S2 in the resting state (from the cryo-EM structures[20,21]); and (ii) the distance change of S4 Trp from surrounding helices upon voltage-dependent activation, determined by DEPET (Supplementary Table 3a, δ column), what are the coordinates of the S4 Trp $C_\alpha$ in the active conformation of the BK channel VSD? The distributions of the S4 Trp $C_\alpha$ coordinates are shown in Supplementary Figure 10.

**DEPET in peptide length standards**. Polyproline-based synthetic peptides of the general formula Cys-(Pro)$_n$-Trp with $n = 1, 2, 3, 4$ or 7 were used as length standards to evaluate the accuracy of DEPET: The C-terminal Trp will quench the N-terminal fluorophore depending on the separated distance (provided by the rigid polyproline chain) according to the FDQ functions. That is:

$$1 - \frac{F_{X,\text{conj}}}{F_{X,\text{free}}} \propto \text{FDQ}_{\alpha\alpha}(X, d_{\alpha\alpha}) \quad (22)$$

where $F_{X,\text{free}}$ is the fluorescence of free (unconjugated) fluorophore $X$ and $F_{X,\text{conj}}$ is the fluorescence of fluorophore $X$ conjugated to a peptide; therefore the left part of the equation is a measure of intramolecular quenching. FDQ($X,d$) is the fluorophore $X$/Trp quenching probability as a function of separating distance, $d$, calculated above.

Each peptide was prepared in lyophilized, 0.5 mg aliquots (Biomatik). Each aliquot was dissolved to 22.5 mM in peptide buffer (50 mM NaHCO$_3$ and 30% acetonitrile, ACN). Each TMRM stock (100 mM in DMSO) was diluted to a 2.25 mM pre-stock in peptide buffer and vortexed vigorously. Peptide and fluorophore were mixed to final concentrations of 50 or 150 μM (peptide) and 3, 7.5, 25 or 50 μM (fluorophore); following thorough mixing, the TMRM fluorophores were allowed to conjugate with the peptide Cys while incubating at room temperature for 1 h. The peptide/dye mixture was diluted 125-fold in peptide buffer before chromatography. We found that reusing peptide aliquots resulted in significantly less peptide labeling efficiency, so only fresh aliquots were used for each preparation. Addition of tris(2-carboxyethyl)

phosphine (TCEP) antioxidant resulted in the elution of multiple fluorophore bands/species, so it was excluded from the labeling protocol.

To measure intramolecular quenching in the fluorophore/peptide conjugates, we used high-performance liquid chromatography (HPLC; controller: Shimadzu Prominence UPLC CBM-20A) followed by absorbance and fluorescence measurements. Each sample was injected (10 μl; autosampler/injector: Shimadzu SIL20AC-HT) into a reversed-phase chromatography column (Shimadzu C18 3 μm, 50 × 4.6 mm) and run at a flow rate of 500 μl/min (two Shimadzu LC-20AD pumps) with the following gradient (solvent A: 50 mM tetraethylammonium acetate (TEAAc); solvent B: ACN): 0 min: 10% B; 1.7 min: 10% B; 28.7 min: 37% B. Each gradient run was followed by two column volumes of rinsing with 100% ACN, followed by two column volumes of 20% ACN. The column output was analyzed by in-line fluorescence ($\lambda_\text{ex} = 550$ nm; $\lambda_\text{em} = 575$ nm; Shimadzu Prominence RF-20Axs) and absorbance (189–800 nm; Shimadzu SPD-M20A) measurements (Fig. 6a).

Absorbance peaks at 550 nm provided a measure of TMRM fluorophore concentration, either free or peptide-conjugated. By including samples of fluorophore without peptide in the runs, it was possible to discern between free fluorophore and peptide-conjugated fluorophore elution peaks. Intramolecular quenching was calculated from the ratio of conjugated vs free TMRM, normalized by their absorbance at 550 nm:

$$1 - \frac{\frac{F_{X,\text{conj}}}{A_{X,\text{conj}}}}{\frac{F_{X,\text{free}}}{A_{X,\text{free}}}} = \text{FDQ}_{\alpha\alpha}(X, d_{\alpha\alpha})c \quad (23)$$

As in DEPET, converging to a solution for $d_{\alpha\alpha}$ (the distance separating the labeled Cys $C_\alpha$ and the Trp $C_\alpha$ atoms in the peptide) and coefficient $c$ requires the simultaneous fitting of multiple fluorophores: in this case, different TMRM fluorophores being intramolecularly quenched when conjugated on the same polyproline peptide (Fig. 6b; fitting parameters in Supplementary Table 4).

The nominal lengths of the peptides were calculated by measuring the Cys $C_a$–Trp $C_a$ distance in polyproline peptides designed in UCSF Chimera[48] using a type II helix structure (proline backbone dihedral angles: $\varphi = -75°$; $\psi = 150°$).

## Data availability

Data are available from the authors upon reasonable request.

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

## Acknowledgements

We are grateful to Drs. Eduardo Perozo, Alan Neely, Susy Kohout, Michela Ottolia and Duncan E. Scott, and members of the Olcese laboratory, for discussions on this work. This work was supported by the NIH/NINDS grant R21NS101734 (R.O.), a NIH/NIGMS Instrument Supplement to R01GM78844 (J.A.), an American Heart Association (National Center) Scientist Development Grant 14SDG20300018 (A.P.) and the Knut and Alice Wallenberg Foundation (A.P.).

## Author contributions

A.P. performed experiments; A.P. and R.O. designed research; A.P. and K.W. developed analysis tools; T.A. and J.A. contributed experimental materials; A.P. and R.O. wrote the manuscript.

## Additional information

**Competing interests:** The authors declare no competing interests.

