## [Peer Review File · Nature Communications]

Reviewers' Comments:

Reviewer #1:

Remarks to the Author:

In this study, Pantazis et. al. describe the application of photoinduced electron transfer to map structural changes in the BK channel. This method is an alternative to FRET based methods and provides distances between the fluorophore attachment site and introduced tryptophans, which serve as a contact quencher. The main modification is that the distances are estimated by measuring the extent of quenching as the linker length between the fluorophore and its attachment site is varied. The idea of using electron transfer quenching as a molecular ruler is not new (see Zhu et. al. Biophysical Journal (2005) 89(5):L37-9). In another study, Jarecki et. al. (Biophys J. (2013) 105: 2724) have used variable linker lengths to determine distances between the probe attachment site and fluorophore. These pioneering studies should be cited and discussed because the principle described here is essentially similar.

The main novelty of this study is the application of this method to BK channels. Using the recently solved structures of the channel in closed and open conformation, they were able to compare the distances between Trp203 in S3-S4 loop with that of Cys136 on top of S1 transmembrane helix. Their data shows that the distance between these sites increases upon channel activation consistent with the available structures. These measurements further serve to validate the distance measurements obtained using the DEPET method. These distances were also calibrated with stiff polyproline peptides.

Major concern:

1. Figure 2a shows that the fluorescence signal changes as the length of the fluorophore and its attachment site is modified but the size of the fluorescence signals typically depend on many factors such as the extent of labeling and number of channels. This contributes to large oocyte to oocyte variation in absolute signals. How do the authors normalize for expression or oocyte to oocyte variability. Typically, the extent of quenching is measured in the same sample but in this study the authors are comparing between different oocytes which will introduce large variability.

Other concerns:

1. Please discuss whether quenching by TRP is dependent on the orientation of the planer aromatic group.
2. State dependence of quenching was measured for only one pair of residues. It is not clear based on a single measurement whether this method is robust enough to predict distances in a protein environment where the fluorophore and quencher have fewer degrees of freedom.
3. Typo first sentence, 2nd paragraph in Page 2.

Reviewer #2:

Remarks to the Author:

The manuscript by Pantazis et al. presents an improved method to measure small distances in proteins. The method is based on an available approach of photo-induced electron transfer (PET) between a fluorophore and a tryptophan residue (Trp) and subsequent quenching of the emitted fluorescence. PET generally occurs when the pair is within a 10 Angstrom distance. The improvements presented in this work have to do mainly with a more rigorous analysis and incorporation of the dynamics of fluorophore and Trp. The authors use molecular dynamics to generate a position function

for both members of the PET pair and then produce a function that encodes the probability of quenching as a function of distance. In this way, the resolution of the method can be increased and distance in the order of a few Angstroms can be measured. Of note, the authors have validated the approach using synthetic rigid peptides of known structure and distances, and show that the measured distances correlate very well with the expected, structure based distances.

The work is novel and presents an improved fluorescence approach applicable to several scientific problems that should appeal to many areas of biophysics. The data has been properly analyzed and the statistical and other quantitative methods are appropriate.

In general the paper is well written, although I do think that a traditional introduction, methods, results, etc. structure would improve the presentation.

My main concern is that the method relies in labeling of an extracellular cysteine with a cysteine-reactive fluorophore and that an implicit assumption of the DPET analysis is that all cysteines are labeled. Is there a way to estimate the fraction of labeled and unlabeled cysteines? Also, four cysteines and TRP residues are present in the channel, what is the effect of the labeling stoichiometry on PET? In other words, given the distances involved, is it possible that PET might happen between one fluorophore and more than one Trp residue?

When the authors calculate the FDQ function for the peptides, it's not clear if the MD was done on the whole fluorophore-peptide-Trp system, please clarify.

In the absence of the Trp203, the signals still show a small quenching effect and for some fluorophores some dequenching is visible, is this solvent-dependent quenching or due to other kind of interactions? Is this non-tryptophan dependent fraction of the signal taken into account on the calculations?

Reviewer #3:

Remarks to the Author:

Pantazis and co-authors describe an approach to monitor distances within proteins by monitoring the quenching of a fluorophore by a nearby tryptophan residue, and the use of this approach to study conformational changes in the large conductance potassium (BK) channel. Although the authors present interesting data, there are several serious problems with the manuscript.

First, this is not a new optical approach (as is claimed in the Abstract), but rather is simply some modifications to an established method that uses fluorescence quenching by tryptophans to glean distance information within biomolecules (see for example Refs 10-14, cited in the paper). Since it is not a novel approach, there is no need for the authors to propose a new term (DEPET, or distance-encoding photoinduced electron transfer) for this method.

Second, it should be noted that a PET quenching mechanism has not even been established as the mechanism for why the tryptophan causes the fluorescence quenching of the probe in this paper. This is an important point - PET is a specific process that differs from collisional quenching or Dexter electron transfer mechanisms, and does not necessarily require contact. The authors make some confusing statements in this regard, such as "The Trp indole side-chain, the agent of collisional quenching,..." in pg. 4 the Supplementary Information.

Third, if one assumes that PET is the underlying quenching mechanism, then a number of caveats exist. The efficiency of PET does depend on distance, but it is also very sensitive to other factors,

including the polarity of the surrounding media, and the relative orientation of the probe vs. quencher (so called edge-to-edge effect), etc. Such factors can vary greatly depending on given protein structure being analyzed, and of course the situation gets more and more complicated and harder to model as larger and larger probes are used. Steric hindrance that limit the accessibility of the tryptophan to the probe can also alter the quenching efficiency.

For these reasons, one cannot assume that the analysis presented here is a general method for determining distances between a fluorophore and tryptophan to angstrom resolution, nor is it correct to assume that data from studies of PET in polyproline helices reliably reflect the situation on a more complex protein structure. Note that even the use of polyproline as a distance calibration tool is controversial (see for example Doose, Neuweiler, Barsch, and Sauer "Probing polyproline structure and dynamics by photoinduced electron transfer provides evidence for deviations from a regular polyproline type II helix". PNAS 104: 17400–17405 (2007)). Although someday it may be possible to extract more precise distance information from PET data for quenching pairs on proteins, the current paper has not established shown this.

It is important to note that this paper does report some good work. The authors present some improvements to the fluorescence quenching by tryptophan method, including the use of molecular dynamics calculations to help analyze the data, and the use of fluorophores of increasing linker-lengths to better estimate distances between the tryptophan and fluorophore. The conformational changes detected in BK are also interesting and impressive. The mathematical models used to help analyze the data and develop a quenching probability function are interesting.

However, because of these issues (difficulty converting PET based data into distances, the size of the probes used, the degrees of freedom introduced by various length linkers, and the reliance on extrapolating data based on polyproline studies to analyze protein structures), it is simply not correct for the authors to make claims like "The accuracy of DEPET is $\sim 1.3 \text{ \AA}$ " (legend of Figure 6).

In summary, while this paper presents some interesting results, due to the lack of novelty about the underlying method, as well as the caveats discussed above, the paper does not warrant publication in Nature Communications.

Reviewer #1

Thank you for evaluating our work and for bringing up important comments and concerns. We have performed new experiments and revised our manuscript based on your comments. We have one objection, on your assessment of the novelty of our work being diminished by previous investigations using variable calipers or collisional quenching. Please find our response to your comments (cited in *blue italics*), below.

The idea of using electron transfer quenching as a molecular ruler is not new (see Zhu et. al. Biophysical Journal (2005) 89(5):L37-9). In another study, Jarecki et. al. (Biophys J. (2013) 105: 2724) have used variable linker lengths to determine distances between the probe attachment site and fluorophore. These pioneering studies should be cited and discussed because the principle described here is essentially similar.

Thank you for bringing up these remarkable studies. We initially focused our background references on Trp-dependent quenching. Of course, we have acknowledged these works in our revised manuscript (Refs.10-11).

We should note that, while electron transfer quenching has been previously used as a molecular ruler, our DEPET approach is unique as it provides (i) relative side-chain orientations and (ii) demonstrated capability of measuring sub-nanometer distances (~0.6nm, compared with ~2.5nm in Jarecki *et al.*: a significant difference of scale for protein structure and function). In addition, the continuous analytical fluorescence/distance (FDQ) functions, facilitate DEPET studies using commercially-available reagents, similar to FRET-based methods.

So while the proposed studies are indeed exemplary works, they do not subtract from the novel achievements made possible by DEPET. Our view is that DEPET takes advantage of an established photochemical reaction (PET) to bring truly new technical innovations and experimental capabilities.

The main novelty of this study is the application of this method to BK channels.

We respectfully, but firmly, disagree with this statement. The quantification of distance- and orientation-dependent quenching afforded by the FDQ functions, which in turn allows the near-Angstrom resolution of protein movements in a fully functional protein (voltage-responsive, conducting ion channel) is the crux of our work's novelty and significance.

Our laboratory has accumulated significant expertise on BK channels, which allowed us to evaluate this new approach in this particular protein, but the theoretical and experimental innovations brought forward in this work are applicable well beyond BK channels.

1. Figure 2a shows that the fluorescence signal changes as the length of the fluorophore and its attachment site is modified but the size of the fluorescence signals typically depend on many factors such as the extent of labeling and number of channels. This contributes to large oocyte to oocyte variation in absolute signals. How do the authors normalize for expression or oocyte to oocyte variability. Typically, the extent of quenching is measured in the same sample but in this study the authors are comparing between different oocytes which will introduce large variability.

Thank you for bringing up this important concern. We incorporate several strategies that both effectively minimize oocyte variability, and account for its effect on the estimated distances:

- 1) All experiments are performed in the same batch of oocyte preparation; that is, all oocytes come from the same frog, and exhibit uniform animal pole pigmentation.
- 2) We have optimized our oocyte labeling protocol to label most channels in the membrane. We found that incubating between 30' and 1 h (on ice) made no difference in the measured ΔF ; we use the 1 hr incubation to ensure maximal labeling. Secondly, we label on ice and use low [fluorophore] (2 μM), to limit background labeling. This step also ensures that our instrumentation works within the same, linear range. Finally, we use reducing conditions (DTT) up to the moment of labeling to ensure that most Cys are available for labeling. We previously neglected to mention this step in our Methods section—we have now corrected this.

- 3) We only measure the ΔF signal, which by definition only arises from labeled channels undergoing conformational changes. Since we can work on conducting ion channels, we normalized the ΔF by the maximal conductance (**Figure S3**), to remove the effect of the (small) variability in the number of channels expressed by oocytes that day.
- 4) Implementing resampling statistics (bootstrapping) in the DEPET analysis workflow to produce distributions of distance solutions effectively translates data variability to a distance confidence interval (**Fig.4**).

That is, we take both experimental (#1-3) and statistical (#4) measures to limit and account for the influence of signal variability as we are comparing different oocytes.

As evidence to that, we have performed a new set of experiments (label at position 136, as previously, but different batch of oocytes) as part of a control experiment requested by Reviewer #2; the $\Delta F/G$ that we measure is comparable with that in original experiments even though it was acquired from a different batch of oocytes, and more importantly, the ΔF proportion from different TMRM fluorophores is very similar (please compare **Fig.S3g** with new **Fig.S4e**).

Ultimately, “the proof is in the pudding”: our distance and orientation measurements of the BK channel VSD are comparable with those in the cryoEM structure (please see revised **Fig.5**).

1. Please discuss whether quenching by TRP is dependent on the orientation of the planar aromatic group.

We have expanded our discussion on the photochemistry of Trp-mediated quenching (p.4 of revised manuscript)

In steady-state measurements (as in our work), Trp quenching can be simplified to a contact process irrespective of orientation. Time-resolved fluorescence measurements have revealed that Trp/fluorophore PET has both a dynamic (collisional) and static (complex formation) component, the latter arising from stacking of the aromatic groups (Doose *et al.*, (2005) ChemPhysChem 6:2277-85)

A simplifying (but necessary) assumption of our FDQ functions is that both interaction modes are jointly calculated as “contact” (probability of Trp and fluorophore to occupy the same spatial interval). This assumption is reasonable because both contact mechanisms act in the nanosecond time-scale, well below the temporal sensitivity of our steady-state fluorescence measurements, and both types of collision are effective in mediating quenching (Doose *et al.*).

Most importantly, the FDQ functions are very accurate in gauging polyproline lengths, as well as structural features of the BK channel, justifying our usage of a contact-probability-based quenching function (irrespective of molecular orientation).

We are wary that this stands for TMRM/Trp quenching; other fluorophores / quenchers may exhibit quenching properties where a simple contact-quenching-based function will not suffice to account for the data. We have updated our discussion of the polyproline experiment to express that this methodology should be used to evaluate the goodness of the FDQ functions of different quencher / probe combinations.

2. State dependence of quenching was measured for only one pair of residues. It is not clear based on a single measurement whether this method is robust enough to predict distances in a protein environment where the fluorophore and quencher have fewer degrees of freedom.

We agree. We are including in this work measurements from two more labelling positions in the BK channel, on VSD helices S0 (new **Fig.S5**) and S2 (new **Fig.S6**). Revised **Figs. 4 & 5** include the new results. The new distances are also in good agreement with the cryo-EM structure of the molluscan BK channel. Notably, we estimate that the S4 Trp side-chain points towards S0 and away from S1 and S2 (revised **Fig.5**). This is important not only because it agrees with the cryo-EM structures from the MacKinnon laboratory, but also because these

data position helix S4 at the center of the VSD, and non-conserved helix S0 at the periphery, in agreement with previous biochemical studies in murine BK channels (Liu...Karlin (2010) *J Gen Physiol* 135(5):449-59).

Last but not least, acquiring distance information from three positions surrounding S4 allowed us to combine the DEPET distance data with the cryo-EM atomi coordinates using trilateration, to determine the position of S4 in the active conformation of the BK VSD (new **Fig.S10**).

3. Typo first sentence, 2nd paragraph in Page 2.

Thank you. We have changed "In lieu or" to "In lieu of".

Reviewer #2

We are grateful for your appraisal of our work, and for raising important questions. Your critique motivated us to perform additional control experiments. Please find our response to your comments (cited in *blue italics*), below.

The work is novel and presents an improved fluorescence approach applicable to several scientific problems that should appeal to many areas of biophysics. The data has been properly analyzed and the statistical and other quantitative methods are appropriate. In general the paper is well written, although I do think that a traditional introduction, methods, results, etc. structure would improve the presentation.

Thank you for positive appraisal of our work. We changed the format to a more traditional layout.

My main concern is that the method relies in labeling of an extracellular cysteine with a cysteine-reactive fluorophore and that an implicit assumption of the DPET analysis is that all cysteines are labeled. Is there a way to estimate the fraction of labeled and unlabeled cysteines?

This (and the following) are both important concerns that we have considered, and we have updated our methods section to clarify our strategies to address them.

Our fluorescence labeling protocol has been optimized to maximize fluorophore association: (i) we keep the oocytes in reducing conditions (DTT) to maintain the reactive availability of extracellular cysteines; (ii) after thorough washing of the DTT, we immediately incubate for a long time (1hr), so that the maleimides will have enough time to react with available Cys. Please note, we previously neglected to mention the DTT step in our Methods section; we have now corrected this.

Since all fluorophores conjugate Cys via a maleimide moiety, they are expected to be equally reactive. In evidence of this, and our protocol's efficacy, (i) the oocyte background labeling is consistent among the fluorophores; (ii) repeating the experiment in oocytes from a different frog resulted in similar $\Delta F/G$ proportions (compare **Fig.S3g** with new **Fig.S4e**) and (iii) no significant increase in labeling is observed after 30 minutes, so we know we are using maximal labeling conditions.

Also, four cysteines and TRP residues are present in the channel, what is the effect of the labeling stoichiometry on PET? In other words, given the distances involved, is it possible that PET might happen between one fluorophore and more than one Trp residue?

This is an important concern for an optical study attempting to measure distance. We have performed additional control experiments to ensure that the fluorophores do not interact with the Trp residues of neighboring subunits, either on the same channel or in adjacent channels. We have included these results in the new **Fig.S4**.

Specifically, we co-expressed BK subunits including an S1 Cys (N136C) and no S4 Trp (W203V), and subunits without the S1 Cys (N136), but with the S4 Trp (W203). Both constructs are in our extracellular-Cys-less background.

In these conditions, observed ΔF were very low, even lower than the residual signal from N136C-W203V channels. If the S4 Trp of adjacent channels and subunits were interacting with the fluorophore, we would expect to see an increased signal.

This result is not surprising, since the range of fluorophore quenching hardly exceeds the diameter of the VSD (new **Fig.S8**). But of course, it is a very important control that supports our premise that the fluorophore:Trp stoichiometry is indeed 1:1 and no inter-subunit quenching takes place.

When the authors calculate the FDQ function for the peptides, it's not clear if the MD was done on the whole fluorophore-peptide-Trp system, please clarify.

We apologize for the lack of clarity. All FDQ functions were constructed by combining the simulations of fluorophore and Trp in isolation (**Fig.3a,b**). We opted to analyse the peptide results in this way because we wanted to evaluate the goodness of the FDQ functions—so it was imperative to estimate the peptide lengths in the same way that we measure distances in proteins.

In the absence of the Trp203, the signals still show a small quenching effect and for some fluorophores some dequenching is visible, is this solvent-dependent quenching or due to other kind of interactions? Is this non-tryptophan dependent fraction of the signal taken into account on the calculations?

We did not originally take the residual ΔF into account, as it is a small fraction of the overall signal. To make sure, we redid our analysis to evaluate its effects, and we present the results of this analysis, below (**Fig.R1**).

We bootstrapped the ΔF signals from the N136C-W203V channels and subtracted them from the bootstrapped samples of the N136C channels, before fitting with FDQ functions. The distance results are very close to those without the subtraction. However, there is a possibility that the ΔF signals from W203V channels do not combine linearly with the Trp-dependent quenching. For this reason, we favor an approach where labeling/quenching positions are chosen that produce significant Trp-dependent ΔF , and analyzing without subtraction.

As to the source of the residual signal, it is a very good question. In the absence of Trp203, the protein still undergoes voltage-dependent movements, and these are reported as residual signals, which may well arise from the differential interaction of fluorophores with the solvent or another quenching group. The important thing is that these signals are a small fraction of the signal in the presence of Trp (or, when the signal with Trp is very small, the signal without Trp is the same).

There were combinations of labeling position and fluorophore that yielded strong signals in the absence of Trp. We have excluded these from our analysis.

Figure R1

Fig. R1: Subtracting the component of Trp-independent ΔF does not significantly alter DEPET distance estimates. (a) Conductance-normalized, voltage-dependent fluorescence change ($\Delta F/G$) from BK channels labeled with TMRM fluorophores of different length in position 136 (S1) helix, in the presence (\circ , left) or absence (\square , right) of W203 in helix S4 (W203V). This is the same data shown in **Fig.S3g**. Representative experiments in **Fig.2a,b**. (b) The samples were bootstrapped 10,000 times, as per usual in DEPET analysis, and are displayed as bins with variable density. Data from W203-less channels were also bootstrapped and subtracted from the data from channels with W203. (c) The subtracted data were simultaneously fit to $FDQ(\alpha\alpha)$ and $FDQ(\alpha\beta)$ functions to determine the distance between the C136 C α atom, and the W203 C α and C β atoms, as in **Fig.4d-f**. The distances were very similar to those without subtraction, reported in the table, below:

	C136 C α — W203 C α distance (Å)		C136 C α — W203 C β distance (Å)	
	Resting state	Active state	Resting state	Active state
Without subtraction (Fig.4d-f, Table S3)	8.0 [7.3,8.8]	14.2 [13.5,15.1]	9.1 [8.5,10.0]	15.7 [14.9,16.6]
With subtraction (this figure)	8.5 [7.9,9.0]	14.7 [14.2,15.1]	9.7 [9.0,10.2]	16.2 [15.6,16.7]

Reviewer #3

Thank you taking time to review and evaluate our work, and for sharing your concerns, which prompted us to more clearly explain the premise of this new approach. Please find our response to all your comments (cited in *blue italics*), below.

First, this is not a new optical approach (as is claimed in the Abstract), but rather is simply some modifications to an established method that uses fluorescence quenching by tryptophans to glean distance information within biomolecules (see for example Refs 10-14, cited in the paper). Since it is not a novel approach, there is no need for the authors to propose a new term (DEPET, or distance-encoding photoinduced electron transfer) for this method.

We respectfully disagree with your suggestion, that DEPET is not new. No work so far (including that in Refs 10-14) describes an optical method capable of resolving sub-nanometer distances and side-chain orientations, in a state-dependent manner, under physiologically-relevant conditions.

Certainly, DEPET development has benefited from rigorous studies on the photochemical interaction of Trp and small organic fluorophores. However, the FDQ functions, which link fluorescence quenching to a distance estimate are completely new and effectively allow the application of DEPET from peptides to an intact, conducting ion channel in a cell membrane. By analogy, cryoEM is based on the principles of EM, but it encompasses additional experimental, theoretical and analytical methods, making it a distinct approach.

There are no examples of quantified distance and orientation resolution in the works that we have cited, and as such do not subtract from the novelty of our work.

Ref. 10: Mansoor, S. E., et al.: This work does not quantifiably measure distances or orientations, but they use a short fluorophore and Trp to determine whether two sites on the T4L protein are within range to allow collisional quenching.

Ref.11: Marme, N., et al.: This work characterized the photochemical properties of quenching of various fluorophores by Trp and other residues, but does not propose a means to extract distance or orientation measurements.

Ref.12: Islas, L. D. & Zagotta, W. N. This work is the first (to our knowledge) to use Trp residues as local fluorescence quenchers in an ion channel. They report ligand-dependent conformational changes, but do not quantify distances or orientations in the resting or active states of the channel.

Ref.13: Mansoor, S. E., et al. In this work, the authors use spectroscopically disparate fluorophores of different length to evaluate the distance of assigned loci in T4L. While they could observe distance-dependent quenching, they could not achieve a quantitative distance or orientation measurement. We overcame this hurdle by developing the FDQ functions.

Ref. 14: Pantazis, A., et al. This is the first work from our laboratory to use Trp-dependent quenching of a fluorophore in the BK channel. As in Islas & Zagotta, we only used one fluorophore, so we could simply conclude that two transmembrane helices in the BK voltage-sensing domain (S0 and S4) move apart upon membrane depolarization: a straightforward structural interpretation, but again, without quantification.

Second, it should be noted that a PET quenching mechanism has not even been established as the mechanism for why the tryptophan causes the fluorescence quenching of the probe in this paper. This is an important point - PET is a specific process that differs from collisional quenching or Dexter electron transfer mechanisms, and does not necessarily require contact. The authors make some confusing statements in this regard, such as "The Trp indole sidechain, the agent of collisional quenching,..." in pg. 4 the Supplementary Information

Thank you for prompting us to clarify the physical chemistry underlying DEPET signals. We agree that "collisional" is the wrong term to use because it describes only one component of PET quenching by Trp. We

have removed all previous instances of “collisional” from the manuscript and expanded on the photochemistry of PET quenching in the main article.

Our understanding, which stems from the seminal photochemical investigations of the Sauer group, is that PET is the principal mechanism by which Trp quenches small organic fluorophores, as it has the most readily oxidizable side-chain of all amino acids (Doose *et al.*, 2005 *ChemPhysChem*).

Dexter transfer and FRET can be ruled out, since both processes require spectral overlap; our TMRM fluorophores are excited by 530nm light (green) and emit at ~565nm (orange), well beyond the near-UV absorption and emission range of Trp.

Doose *et al.* have shown that PET has both a dynamic (collisional) and static (complex formation) component. Both require van der Waals overlap, *i.e.*, contact, as it is an e^- transfer reaction. Our steady-state, ensemble fluorescence measurements, cannot discriminate between the two Trp-contact-mediated modes of PET, as both occur within a nanosecond timescale; accordingly, the FDQ functions correlate the probability of contact-based quenching with distance, satisfying the central requirement of PET (that is, contact between the fluorophore and the Trp side-chain).

Importantly, the FDQ functions are very accurate in gauging polyproline lengths, as well as structural features of the BK channel, validating our fluorescence-based distance measurements with a contact-based quenching mechanism. We have further supported our approach with additional control experiments testing the possibility of inter-subunit quenching (new **Fig.S4**) as well as performed distance and orientation measurements from two additional positions on the BK channel voltage-sensing domain (new **Figs. S5** and **S6**).

Third, if one assumes that PET is the underlying quenching mechanism, then a number of caveats exist. The efficiency of PET does depend on distance, but it is also very sensitive to other factors, including the polarity of the surrounding media, and the relative orientation of the probe vs. quencher (so called edge-to-edge effect), etc. Such factors can vary greatly depending on given protein structure being analyzed, and of course the situation gets more and more complicated and harder to model as larger and larger probes are used. Steric hindrance that limit the accessibility of the tryptophan to the probe can also alter the quenching efficiency. For these reasons, one cannot assume that the analysis presented here is a general method for determining distances between a fluorophore and tryptophan to angstrom resolution, nor is it correct to assume that data from studies of PET in polyproline helices reliably reflect the situation on a more complex protein structure.

These are important concerns that we have previously considered and excluded from our experimental design. Ultimately, as our results show, any contribution of the suggested caveats is insufficient to limit the measurement of (i) polyproline peptides or (ii) BK channel voltage-sensor movements; therefore, our approach, while not without limitations or simplifying assumptions, is suitable for structural biological investigations. A more detailed rebuttal of the proposed caveats is below:

All fluorescence signals arise from the interaction of fluorophore and Trp, as shown in Trp-less (W203V) channels; changes in media polarity do not sufficiently affect the fluorescence signal in the labeling position used (small ΔF from W203V channels; **Figs. 2, S3g, S4, S5, S6**). Labeling other positions can result in more Trp-independent fluorescence change, and they have been excluded from subsequent analysis.

Different physical Trp/fluorophore interactions (collisional / complex formation), and by extension the orientation of indole/xanthene contact, occur well below our temporal resolution limit, *i.e.*, they are all reported as steady-state fluorescence quenching. Since both require contact for the e^- transfer to take place, our technique is theoretically valid.

We specifically limit our use to small probes because (i) their all-atom simulations produce reliable results and (ii) to measure short distances and distance changes. We do acknowledge in our discussion, that contact-based quenching is better for sub- or near-nanometer distances, while FRET is more suitable to measure longer

distances, even though longer tethered fluorescence quenchers (Jarecki *et al.*, *Biophys J* 2013) or K⁺ channel blockers (Blaustein *et al.*, *Nat Struct Biol* 2000) have been previously used.

Steric hindrance may limit fluorophore/Trp interactions so the choice of labeling / Trp positions is important. Positions that yield no ΔF signal, or that produce ΔF in the absence of Trp, were excluded. Like traditional FRET-based approaches, choosing suitable protein positions to label is a critical first step. Thankfully, there is no lack of extracellularly-accessible positions to label in a transmembrane protein such as an ion channel; and recent developments in patch-clamp fluorometry and fluorescent unnatural amino-acids will expand the applicability of DEPET to previously-inaccessible intracellular domains.

Note that even the use of polyproline as a distance calibration tool is controversial (see for example Doose, Neuweiler, Barsch, and Sauer "Probing polyproline structure and dynamics by photoinduced electron transfer provides evidence for deviations from a regular polyproline type II helix". PNAS 104: 17400–17405 (2007)).

We are familiar with this excellent work, and its subject does not affect our polyproline experiments. A bit of background: Polyproline peptides have been historically used as length standards because Pro residues adopt a trans configuration in polar solvent, resulting in a stiff "type II" helix of reliable length. Rarely, Pro can isomerise to a cis configuration (usually encountered in non-polar solvent), that can introduce a kink in a long polyproline type-II helix. In this work, Doose and colleagues examine trans-cis isomerizations in polyproline peptides, and how they can make polyproline peptides assume a shorter effective length.

We have no evidence that trans-cis Pro isomerization significantly affected our results. If trans-cis transitions were prevalent, we would expect to measure shorter peptide lengths than expected (evidenced as stronger quenching in the shorter fluorophores than anticipated), which is not the case: The average error of our polyproline measurements was -0.22\AA ; that is, our distance estimates were overall slightly short, but hardly evident of a systematic length underestimation.

This is not surprising: trans-cis prolyl isomerization is very rare under polar solvent conditions; since it occurs stochastically, it is more likely to affect measurements in longer polyproline peptides. Our peptides were relatively short (1,2,3,4 or 7 Pro), which alleviates this concern. By comparison, Stryer and Haugland used up to 12-proline-long peptides to evaluate FRET (PNAS, 1967); Jarecki and colleagues (*Biophys J*, 2013) used 6-10 prolines to calibrate their molecular calipers.

On this note, we must emphasize that we only used polyprolines to evaluate FDQ accuracy, and not to calibrate our measurements; that is, our findings on the peptides do not at all affect the capability of DEPET to measure distances in proteins. While not perfect, polyproline chains are the gold standard of nanometer distances and are routinely used to both evaluate and calibrate length measurements.

Although someday it may be possible to extract more precise distance information from PET data [...] [issues regarding] difficulty converting PET-based data into distances

We are puzzled as to why the Reviewer thinks so. While further improvements are possible (e.g., better molecular dynamics simulations, fluorophores with more "contoured" distance profiles), the current FDQ functions (and DEPET implementation) work very well in predicting peptide length standards and the BK channel structure from steady-state fluorescence measurements, offering distance and orientation resolution in the league of crystallographic methods, while allowing for physiologically-relevant experimental conditions.

[issues regarding] the size of the probes used

It is not clear how differently-sized fluorophores would have produced better results. The probes were ideally sized for the measurement of the nanometer distances in the BK channel voltage sensor domain (please see

new **Fig.S8**). In addition, they are commercially available and spectrally identical (**Fig.S1**). The availability of smaller probes and even fluorescent unnatural amino-acids increases the applicability of our method.

[issues regarding] the degrees of freedom introduced by various length linkers

This is certainly taken into account, by the molecular dynamics simulations of the fluorophores (**Figs.3, S7, S9**).

[issues regarding] the reliance on extrapolating data based on polyproline studies to analyze protein structures

We hope there has been no misunderstanding: we certainly did not extrapolate polyproline studies to protein analysis, nor do we rely on them to calibrate protein structure measurements.

We developed the theoretical framework (FDQ functions) *de novo*, based on the contact-based nature of Trp-mediated quenching, and evaluated it in both polyproline peptides (of reasonably predictable structure) and the BK channel protein (structure resolved by cryoEM). In both evaluations, the FDQ functions gave impressively good results in terms of accuracy and orientation.

it is simply not correct for the authors to make claims like “The accuracy of DEPET is ~1.3 Å”

We have moderated the language in the title of this section, and clarified the text.

We do believe that the polyproline study reflects a “baseline” measure for DEPET accuracy because, assuming the expected lengths of the polyproline peptides are fully accurate, the accuracy (*i.e.*, lack of error) of the distance measurement relies solely on the goodness of the FDQ functions.

We cannot use the BK channel cryoEM structure to evaluate DEPET accuracy in a quantified manner, because (i) the cryoEM structure was resolved in a molluscan BK channel (DEPET was done on the human protein); (ii) the resolution of the cryoEM structure coordinates was 3.5 Å and (iii) the experimental conditions were very different.

We should add that, when working on a labeled membrane protein undergoing conformational change, the implementation of resampling statistics (bootstrapping) in the DEPET analysis workflow allows the determination of a confidence interval for the distance solutions.

In summary, while this paper presents some interesting results, due to the lack of novelty about the underlying method, as well as the caveats discussed above, the paper does not warrant publication in Nature Communications.

The underlying method does involve sufficient novelty.

It does take advantage of the established photochemistry between fluorescent dyes and a contact quencher; and we do use variable-length calipers, as did previous pharmacological and optical investigations. These are fundamental theoretical and practical elements on which any contact-quenching-based approach will be based.

However, the analytical framework to convert PET data into sub-nanometer distances and, most uniquely, relative side-chain orientations, is absolutely new and yields unprecedented experimental results. We have demonstrated its capability, using commercially-available reagents, in both polyproline length standards and in a conducting human channel protein undergoing physiological conformational changes in the membrane of a cell.

In **Fig.5**, we demonstrate how our approach recapitulates, in the sub-nanometer scale, structural features of a membrane protein resolved by cryoEM; in addition, DEPET captured how this structure changes in response to

a natural stimulus (membrane depolarization); and its data came from functional, conducting ion channel proteins expressed in living cell membranes. Surely DEPET qualifies as a new method capable of novel findings. We are grateful for the opportunity to have our work judged by an expert in the field, and we sincerely believe that the additional work presented in the revised manuscript and our clarifications and rebuttals to your concerns will be to your satisfaction.

Reviewers' Comments:

Reviewer #2:

Remarks to the Author:

The authors have satisfactorily answered my concerns and addressed my comments on the manuscript. The new version of the manuscript is much improved and it is easier to read. The method they present is well documented and represents an advance in the available methodologies to measure short distances in membrane proteins in vivo.

Reviewer #4:

Remarks to the Author:

Revs 1 and 3 raise concerns about novelty of the method, calling attention to prior work with Trp quenching. Because of this Rev 1 concludes that the main gain is for the study of BK channels. Rev 3 says this means that the paper is not suitable for the journal.

The authors' response acknowledges that Trp quenching is not new but explains that use of FDQ analysis is and that the payoff for this analysis is novel: unprecedented increase in resolution + critical information on angle.

The agreement of the measurements with the CryoEM structure are impressive. And the estimated resolution of the approach is as well. Experimental concern raised in the review about the limited data set has been dealt with by adding new sites.

This leaves some specific concerns (which are well answered) and the major question about novelty.

In the eyes of this reviewer, the paper is a significant step forward for the field. Although Trp quenching is not new, the technical advance in its application is and this makes all the difference in the world. The results are a remarkable match between frozen structure and average structure in a live dynamic protein in its native membrane environment. Since crystallography and CryoEM are not possible on all proteins, and cannot often match functional state to structural state in an unambiguous manner, it has been a holy grail to have a method that can relate the two.

The concerns raised by the reviews about polypro calibration, suitable fluorophores, generalizability, etc remain relevant. But the correspondence is there, as is the information about protein motion in the live protein and at super-hi resolution. This is remarkable, and it will be appreciated in the field.